# Manipulation of Focal Adhesion Signaling by Pathogenic Microbes

**DOI:** 10.3390/ijms22031358

**Published:** 2021-01-29

**Authors:** Korinn N. Murphy, Amanda J. Brinkworth

**Affiliations:** 1School of Molecular Biosciences, Washington State University, Pullman, WA 99164, USA; korinn.murphy@wsu.edu; 2Department of Pathology and Microbiology, University of Nebraska Medical Center, Omaha, NE 68198, USA

**Keywords:** focal adhesions, pathogenesis, vinculin mimetic, outside–in signaling, host–pathogen interactions, stress fibers, integrin signaling

## Abstract

Focal adhesions (FAs) serve as dynamic signaling hubs within the cell. They connect intracellular actin to the extracellular matrix (ECM) and respond to environmental cues. In doing so, these structures facilitate important processes such as cell–ECM adhesion and migration. Pathogenic microbes often modify the host cell actin cytoskeleton in their pursuit of an ideal replicative niche or during invasion to facilitate uptake. As actin-interfacing structures, FA dynamics are also intimately tied to actin cytoskeletal organization. Indeed, exploitation of FAs is another avenue by which pathogenic microbes ensure their uptake, survival and dissemination. This is often achieved through the secretion of effector proteins which target specific protein components within the FA. Molecular mimicry of the leucine–aspartic acid (LD) motif or vinculin-binding domains (VBDs) commonly found within FA proteins is a common microbial strategy. Other effectors may induce post-translational modifications to FA proteins through the regulation of phosphorylation sites or proteolytic cleavage. In this review, we present an overview of the regulatory mechanisms governing host cell FAs, and provide examples of how pathogenic microbes have evolved to co-opt them to their own advantage. Recent technological advances pose exciting opportunities for delving deeper into the mechanistic details by which pathogenic microbes modify FAs.

## 1. Introduction

The ability to actively manipulate eukaryotic host cells is a hallmark of many pathogenic microbes. Viruses, bacteria, and parasites share the same refined ability to invade host cells and induce complex changes to the environment around them. Many of these changes have a direct bearing on the virulence of a given microbe, or its ability to cause harm to its host during the course of an infection. Indeed, infectious diseases continue to pose a substantial threat to human health, placing an enormous burden on health systems and contributing to over 17 million deaths per year [1]. A central pillar of infectious disease research has focused on understanding how pathogens manipulate host cell dynamics to facilitate disease. In this review, we hope to highlight the growing body of work defining a role for focal adhesion complexes as important cellular structures modified by pathogens to help drive infection. Focal adhesion modulation has emerged as a significant pathogenic mechanism, though it has been less studied historically than the cytoskeletal rearrangement known to occur during pathogenesis.

The cytoskeleton, as the core structural component of cells, is unsurprisingly a key target manipulated by many pathogens. The cytoskeleton encompasses a network of filaments (actin, microtubules, intermediate filaments) along with filament-forming proteins such as septins. This network provides essential structural support to maintain proper positioning of a cell’s shape as well as positioning of its constituent organelles [2]. Several excellent reviews exist that detail the way pathogens restructure the host actin cytoskeleton to facilitate infection [3,4]. Importantly, this manipulation is not restricted to a singular stage of pathogenesis, but rather has been implicated in diverse events in a pathogen’s lifecycle such as invasion, replication and dissemination. For intracellular pathogens, entry into the host cell represents a key occurrence of actin remodeling facilitated by the pathogen. During invasion, the introduction of secreted effector proteins into the host cell is a common mechanism enabling intracellular pathogens to reorganize actin and promote internalization [5]. Cytoskeletal rearrangement has also been linked to pathogen survival and replication following internalization. To maintain an infectious foothold, intracellular pathogens utilize actin remodeling to fulfill a variety of purposes, such as formation of a filamentous cage that lends structural support to bacteria-containing vacuoles, formation of actin tail structures that can propel organisms through the cytoplasm, as well as manipulation of the cytoskeleton’s vesicular trafficking to promote nutrient acquisition. Studies examining pathogen dissemination events tell a similar story. There are several egress strategies routinely utilized by pathogens to exit a host cell where actin rearrangement proves indispensable. Bacteria-driven filopodia as well as extrusion of membrane-bound vacuoles are two prime examples [6]. The breadth of ways in which targeting the cytoskeleton proves beneficial for pathogen survival has been rather well studied in recent years. This central role for cytoskeletal remodeling prompts the question—what other actin-interfacing structures might be co-opted by pathogens to drive cellular infection?

The actin cytoskeleton is anchored to the extracellular matrix (ECM) by dynamic, multi-protein complexes known as focal adhesions (FAs). FAs contain many actin-binding proteins, which is central to their ability to link extracellular, ECM-bound integrin receptors with intracellular actin (Figure 1) Thus, FAs have an intimate relationship to the actin cytoskeleton [7]. Containing over 100 different proteins, adhesions display a notable degree of functional diversity. An array of scaffolding, adaptor and regulatory functions have been assigned to FA proteins. These functions position FAs to serve as key signaling hubs within the cell. Indeed, FA complexes are capable of transmitting a number of environmental cues about the extracellular environment. This sensing allows cells to respond to changes in the chemical or physical properties of their surroundings [8,9].

This collection of FA proteins, together comprising the integrin “adhesome”, contain a variety of functional protein domains. These domains are involved in protein–protein interactions at adhesions, and are responsible for driving protein recruitment and post-translational modifications within the adhesome. FA proteins rely upon these protein–protein interaction domains to facilitate complex signal transduction pathways [10]. For a pathogen seeking to invade a host cell, such signal transduction pathways make ideal targets to induce host cell remodeling. For many pathogens, this takes the form of molecular mimicry, a virulence strategy defined by sequence or structural resemblance between microbial and host molecules [11,12].

A key feature of FAs is their dynamic nature. Turnover of adhesion components occurs at different stages of the adhesion’s lifecycle to facilitate cellular processes such as cell migration. It is now appreciated that newly formed “nascent adhesions” are less stable structures, which may disassemble or mature into more stable structures referred to as “focal complexes”. From there, focal complexes can further mature into larger, elongated “focal adhesions”. The term “fibrillar adhesion” has been used to describe a subset of tensin-rich, elongated adhesions located at the cell center and enriched for matrix components such as fibronectin. Incorporation of additional protein components is characteristic of the FA maturation process [13,14]. This differentiation between transient nascent adhesions and those that mature into longer-lived FA complexes is especially interesting in the context of pathogenesis. Several pathogens have been shown to recruit FA-associated proteins to the site of invasion. These transient associations raise intriguing questions—do such clusters of FA-associated proteins function as a type of “pseudoadhesion” even when disassociated from the basolateral surface of cells, and if so, what is the extent of the signaling transduction that may occur? Additionally, does pathogen subversion of FA proteins during the invasion process have implications for their function elsewhere in the cell? Do these proteins have post-invasion roles in maintaining infection? In this review, we will examine how pathogens can modulate both transient FA complexes as well as stable adhesions within the cell. The necessity for FAs to dynamically assemble and disassemble requires an exquisite level of regulation. Modulation of phosphorylation through the action of kinases and phosphatases, degradation via proteolytic cleavage of adhesion components, autoinhibitory mechanisms and mechanotransduction all play a role in this regulation.

Dysregulation of FA signaling has been implicated in a variety of human disease states. Kindler syndrome is a human genetic disorder characterized by blistering and fragile skin that is caused by impaired FA protein function. Several FA proteins that are essential during embryonic development, such as integrin-linked kinase (ILK), continue to prove vital to the correct functioning of tissues and organs in adults [15]. However, the best-studied association between FA proteins and human disease occurs during cancer. Cancer cells often exhibit altered FA dynamics, which contribute to oncogenic events such as increased cell proliferation, as well as enhanced cell motility [16]. Given this capacity to contribute to human disease, it is no surprise that disrupted FA signaling can also facilitate infectious disease. In this review, we will first summarize the field’s current understanding of FA-regulatory mechanisms and how they facilitate a diverse role for FAs in controlling important cellular functions such as cell adhesion and migration. We hope that this outlook will be helpful in contextualizing the second focus of our review, to define how an array of pathogenic microbes have been shown to subvert FA signaling to facilitate pathogenesis, both during and after host invasion.

## 2. Multiple Regulatory Mechanisms Control Focal Adhesion Dynamics

This review will primarily focus on a subset of the most well-characterized molecular components that comprise a focal adhesion, including the proteins FAK, Src, integrin-linked kinase (ILK), paxillin, p130Cas (Crk-associated substrate), Rap1-interacting adapter molecule (RIAM), talin, vinculin, α-actinin and zyxin. In order to perform their role in signal transduction, FA complexes must adopt the correct multi-protein structural arrangement. For this reason, FA proteins are often classified as either adaptor proteins which facilitate protein interactions or signaling proteins which exhibit enzymatic activity. However, these categorizations are not mutually exclusive, as there are many proteins which have both catalytic activity and protein-binding domains. Kinases (e.g., FAK, Src and ILK) are examples of catalytic proteins, albeit with different substrate specificity. FAK and Src are tyrosine kinases and represent two of the major protein kinases present within FAs [17]. Both are notable for their ability to recruit and phosphorylate proteins within the adhesion [18]. ILK is a serine/threonine protein kinase that regulates protein–protein interactions and is an important partner of the β1 integrin cytoplasmic domain [19]. Adaptor proteins fulfill their function through a variety of protein-binding domains commonly found within FA proteins. Some proteins contain more than one type of domain, which contributes to the varied protein–protein interactions observed at FA sites. The protein domain structures for the proteins discussed within this review, along with a brief summary of common domain functions, are provided in (Figure 2).

RIAM belongs to the class of FA proteins which help regulate the integrin component of the adhesion complex. RIAM fulfills this function by interacting directly with talin and helping localize it to the plasma membrane for integrin engagement [42]. Talin is a core structural adaptor protein which activates integrins by binding to the tail of the β-integrin subunit [43]. Additionally, by binding F-actin, talin directly couples integrins to the cytoskeleton. Talin has been called the “master of integrin adhesion” to highlight how critical the integrin–talin–actin linkage proves to FA growth and stability [44]. Talin contains numerous binding sites for the FA protein vinculin. Due to the actin-binding ability of the adaptor protein vinculin, its recruitment provides further stabilization to the connection between a FA complex and actin [45]. Together, the talin–vinculin interaction helps promote adhesion maturation. Paxillin is a key scaffolding protein at FAs and is responsible for the recruitment of an array of proteins with enzymatic or structural function that facilitate FA signaling [46]. One of these adaptor proteins is p130Cas, whose interaction with the LD1 motif of paxillin has been shown to play a role in its targeting to FAs [47]. In addition to promoting protein–protein interactions at FAs, p130Cas is a Src substrate which can undergo tyrosine phosphorylation to facilitate downstream signaling [48]. The α-actinin family are another group of actin-binding proteins. These proteins exist as anti-parallel dimers whose structure allows them to effectively cross-link actin filaments [49]. As FAs mature, they incorporate additional actin-regulatory proteins. Zyxin, a binding partner of α-actinin, is one such protein which incorporates into the adhesion at later stages of development. Zyxin is enriched along actin filaments, where it can localize to promote stress fiber stabilization or repair [50]. There are examples of pathogenic microbes interrupting the function of each of these FA components. In order to understand how pathogens subvert FA signaling to their own advantage, we will first provide an overview of how normal FA biology is tightly regulated by the cell.

FAs play a well-defined role in a number of physiological processes critical to the cell. These multi-protein complexes are known as key transducers of cell survival signals and as dynamic sensory hubs—triggering cells to proliferate, migrate and differentiate in response to appropriate cues in their microenvironment [51]. Almost half a century of research has gone into discovering how these structures properly exert their function. Clearly, FAs require a high degree of temporal regulation, as their protein components dynamically assemble and disassemble to facilitate cell migration. Additionally, precise spatial regulation is also required, as many FA proteins engage with more than one binding partner within the adhesion [52]. Proper targeting of protein components to the site of the FA is also critical for correct function, as most FA proteins transiently cycle between FA-bound and cytoplasmic fractions. Notably, some FA proteins can also shuttle to the nucleus and have been implicated in the control of gene expression [53]. Given this complexity, how then are FA proteins able to perform their function when and where they are needed? Multiple points of regulation allow FA proteins to perform these multi-faceted roles in the cell (Figure 3). These core regulatory mechanisms are reviewed in the following sections.

### 2.1. ECM Stiffness Sensing of FA Proteins

The fine-tuned ability of FAs to sense and respond to changes in their cellular microenvironment is in large part due to the bi-directional nature of integrin signaling. As integrin-containing complexes, FAs are sensitive both to intracellular “inside–out” signals as well as the “outside–in” signal transduction generated from integrin engagement with the ECM. As heterodimers, integrin receptors contain both an α and a β subunit. While 24 different integrins exist, each with preferential affinity for specific ECM ligands, matrix binding interactions are predominantly facilitated by β1 integrin receptors [54]. The ECM is another source of structural support for the cell, and is comprised of a large array of macromolecules that integrin receptors can bind. Molecules such as proteoglycan, fibronectin, vitronectin, elastin, collagen and laminin can all be found within the ECM protein network. Importantly, the ECM is not a static scaffold, but rather is dynamically remodeled by the cell through the active secretion, deposition or degradation of ECM components [55]. In this manner, diverse biochemical cues can be produced by the unique interactions between different receptors and matrix ligands. The physical properties of the ECM, such as its rigidity and density, can also change in response to the specific molecular composition of the ECM at a given time. These changes have drastic effects on global cell attributes such as cell shape and proliferation. Indeed, several disease states have been linked to altered mechanical properties of the ECM.

FAs are responsive to these changes, which are sensed by integrins. Particular focus has been placed on understanding the role of matrix rigidity in modulating FA biology. The rigidity of the matrix, also discussed in regards to substrate softness or stiffness, is a byproduct of the composition and organization of the ECM, as well as post-translational modifications such as enzymatic cross-linking [56]. Changes in matrix rigidity are sufficient to induce changes to the composition and signaling exhibited by FAs. Studies conducted by Prager-Khoutorsky et al. utilized fibronectin-coated poly(dimethylsiloxane) gels measured at either 5 kPa (compliant) or 2 MPa (rigid) tensile stiffness in order to investigate adhesion dynamics [57]. They observed that growth on the rigid (stiff) surface resulted in an approximate two-fold increase in FA size. Live-cell imaging also indicated that adhesions on the rigid substrate were less dynamic than their counterparts grown on compliant (soft) surfaces. Overall, stiffness resulted in larger and more stable adhesions. Notably, substrate stiffness also modulated cellular morphology, as cells grown on soft surfaces were demonstrated to be generally rounder and less spread out. Cell polarization was also shown to be rigidity dependent, with elongated and polarized cells forming on the stiff substrates. Matrix rigidity has also been demonstrated to play a role in FA maturation by promoting the growth of fibrillar adhesions. In turn, fibrillar adhesions remodel the ECM to induce fibrillogenesis [13]. Many studies in the field rely upon stiffness gradient hydrogels, whose rigidity can be characterized using atomic force microscopy, to assay stiffness-dependent changes to FAs [58].

An interesting question that is emerging is whether matrix stiffness sensing via FA complexes has a bearing on host–pathogen interactions during infection. Bastounis et al. addressed this question in the context of *Listeria monocytogenes* infection of endothelial cells [59]. As noted by the authors, *L. monocytogenes* is an ideal model organism for stiffness gradient studies due to its broad tissue tropism. The ability to infect a wide array of tissue types, with variable surrounding ECM, means the pathogen is likely to encounter natural stiffness gradients during the course of an in vivo infection. Bastounis et al. utilized a polyacrylamide hydrogel model to assay uptake of the bacterium for cells seeded on soft (3 kPa) or stiff (70 kPa) matrices. Bacterial uptake was found to increase with increasing hydrogel stiffness. Next, they investigated the connection between matrix stiffness and FA signaling, by probing the phosphorylation state of the tension-responsive Y397 residue of FAK. Soft matrices exhibited decreased FAK phosphorylation compared to stiff matrices. Additionally, decreased bacterial uptake was observed for cells treated with FAK inhibitors. Conversely, elevating FAK activity through the action of angiotensin II increased the cells susceptibility to infection. To differentiate whether matrix stiffness was exerting an influence at the level of bacterial adherence or bacterial invasion, assays with a GFP-expressing strain of *L. monocytogenes* were utilized in conjunction with antibody-labeling under non-permeabilizing conditions, such that bacteria which were adhered but not internalized by the cell could be identified. They concluded from this experiment that bacterial adherence was the major factor influenced by gradient stiffness, as the invasion efficiency ratio of internalized to total bacteria did not change across matrices. Finally, they identified vimentin as a FAK-responsive host cell receptor that also contributes to *L. monocytogenes* adhesion. A dose-dependent decrease in bacterial uptake was observed when cells were pretreated with the anti-vimentin antibody H-84. Altogether, their findings point to ECM stiffness as an important mediator of *L. monocytogenes* uptake, as well as implicate a role for FAK activity and the host cell receptor vimentin. This study raises intriguing questions about how ECM stiffness may modulate host cell susceptibility to infection for a variety of potential bacterial pathogens. Certainly, it is evidence that the FA signaling induced by ECM stiffness should not be discounted as an important variable in our larger understanding of how host–pathogen interactions drive infection.

### 2.2. Tension Responsiveness of FA Proteins

Cells must be able to respond to mechanical force to perform a variety of routine cellular processes. Through the action of mechanosensitive proteins, mechanical force can be converted by the cell into sophisticated biochemical signalling responses. The cytoskeletal network plays a critical role in this transduction, transmitting mechanical force along filaments such as actin and microtubules [60]. As actin-binding structures, FAs are sensitive to these intracellular mechanical forces, as well as external forces which originate from the ECM. In part, FAs are responsive to changes in substrate rigidity because they are composed of a repertoire of mechanosensitive proteins [61]. This intracellular tension dependence is demonstrated by FA sensitivity to myosin II activity. In response to tension supplied by myosin II, the force-dependent recruitment of proteins such as zyxin and α-actinin occurs, and FA complexes undergo maturation [62]. Treatment with the pharmacological agent blebbistatin, a specific myosin II inhibitor, induces the disassembly of stress fibers and FAs. However, in a scenario where the adhesion is under high tension but stress fiber assembly is absent, FA maturation is no longer induced. Therefore, tension is required for FA growth and maturation, but the contribution of stress fibers as a template for FA growth cannot be discounted [63]. With regards to external forces from the ECM, tension on integrin has been shown to enhance RhoA activation. RhoA-stimulated tension also influences FA maturation.

Several proteins at the adhesome are stretched in response to mechanical force, with varied consequences to their activation state or ability to engage in specific protein–protein interactions. For example, stretching of the talin molecule modulates FA dynamics by exposing additional binding sites for vinculin that were previously buried [64]. The interaction between talin and vinculin stabilizes adhesins, helping facilitate force transduction. However, vinculin is not the only talin-binding FA protein. In fact, the interaction between talin and RIAM is responsible for the initial recruitment of talin to integrins. Interestingly, these two talin-binding proteins, vinculin and RIAM, have been shown to bind talin in a mutually exclusive manner [65]. RIAM is abundant at the plasma membrane. However, due to direct competition for talin-binding sites, vinculin predominates at mature adhesion sites. Vinculin’s ability to displace RIAM at the adhesion site drives the transition from nascent adhesion to a stable adhesion that can transduce force. In this manner, the force-induced domain unfolding of talin stimulates vinculin binding while displacing RIAM. Actomyosin stimulates the sequential displacement of RIAM in favor of vinculin binding to talin [66]. This is a key example of mechanical force altering the structure of a FA protein to induce a biochemical response capable of altering adhesion dynamics. Kumar et al. developed a FRET-based tension sensor in order to study the dynamics of talin tension at FAs [67]. They validated that talin is under tension, and determined that this tension is higher in peripheral as opposed to central adhesions. This observation is consistent with the idea that tension at early adhesion sites is important to promote vinculin binding and actin engagement. They implicated talin’s actin-binding site 2 (ABS2) as the primary mediator of tension, rather than talin’s actin-binding site 3 (ABS3) which is present at the C-terminus. To follow up on these studies, they coupled their FRET imaging to cellular cryo-electron tomography (cryo-ET) in order to investigate talin tension in the context of local actin organization [68]. They found that regions of high talin tension corresponded to highly aligned actin filaments. This is a prime example of how spatial dynamics can dictate the mechanical response of a FA complex.

While talin is a major facilitator of force transmission at adhesions due to its direct binding of both integrin and F-actin, other adhesion proteins have also been shown to be responsive to mechanical force. FAK is one such protein, whose activation is sensitive to local substrate rigidity [69]. FAK-null fibroblasts are impaired in their ability to respond to mechanical force during migration, which manifests as defects in migration speed as well as in directional durotaxis—the ability to migrate from rigid to soft substrates [70]. Additionally, this function has been shown to rely upon phosphorylation at the Y397 autophosphorylation site, as evidenced by studies with an inactive FAK-F397 mutant [71]. Tension-induced FAK activation has also been shown to differ across ECM molecules. Studies utilizing a FRET-based biosensor revealed that FAK activity increased proportional to substrate rigidity for cells adhered to fibronectin, whereas cells grown on collagen I did not exhibit the same dependency. This suggests that FAK’s mechanosensitivity is mediated through a fibronectin–integrin signaling axis [72]. Recent research has sought to clarify whether FAK mechanosensing operates completely downstream of integrins via an indirect mechanism only, or if mechanical force directly activates FAK. Zhou et al. utilized molecular dynamics simulations to approach this question, and observed that mechanical force acting between the basic patch of the FERM domain and the C-terminal kinase domain triggers dissociation that could relieve FAK’s autoinhibitory state [73]. This dissociation of the autoinhibitory FERM domain from the kinase domain promotes FAK activation. Bauer et al. provide further support for the direct activation model and propose that the mechanical force measured at FA sites is sufficient to cause force-induced conformational changes in FAK [74]. Furthermore, they posit that this force on the C-terminus of FAK may be mediated by interaction with the FA proteins paxillin and vinculin.

Mechanical force has also been demonstrated to play a role in the phosphoregulation of the adaptor protein paxillin. Paxillin phosphorylation at Y31 or Y118 is critical for controlling adhesion turnover dynamics, as phosphorylation of paxillin precedes FA disassembly [75]. Conversely, concomitant with force-induced growth of adhesion sites is a decrease in the phosphorylation of paxillin. Additionally, when adhesion strength was challenged in paxillin-deficient cells via the application of high levels of shear force, only the unphosphorylated complemented version of paxillin (Y31F/Y118F) rescued cellular adhesion strength [76]. Vinculin turnover dynamics are also responsive to paxillin’s phosphorylation state. Additional evidence for paxillin’s mechanosensing is its ability to respond to mechanical stress by mobilizing to sites of stress fiber strain. This process depends upon its LIM domains. Once recruited, paxillin can help mediate the repair and stabilization of actin stress fibers. Zyxin is another LIM domain-containing protein that has been shown to possess actin repair function, though it operates independently from paxillin-mediated repair [77]. Zyxin was initially identified as a mechanosensitive protein due to its mobilization from FA sites to actin filaments with the application of cyclic stretch. In the absence of zyxin, actin filaments exposed to cyclic stretch are much thinner, suggesting that zyxin plays a critical role in mechanically induced stress fiber reinforcement and thickening [78]. Furthermore, zyxin accumulates within force-bearing sites. The pharmacological agents Y27632 (Rho-kinase inhibitor) and blebbistatin (myosin II inhibitor) have both been shown to decrease traction force and zyxin accumulation, suggesting a myosin II-mediated mechanism of the force-dependent recruitment of zyxin to FAs [79]. However, there is also evidence that zyxin-dependent stress fiber reinforcement can still occur even in the presence of the Rho-kinase inhibitor. The actin remodeling mechanical response of zyxin is also dependent upon its binding partners α-actinin and VASP [80].

Zyxin interacts directly with a number of FAs that are also force responsive, including α-actinin and p130Cas. Actinins play a defined role in cross-linking actin filaments. In addition, actinins have also been implicated in the FA maturation process. α-actinin facilitates force transduction between integrins and actin within nascent FAs, thereby triggering adhesion maturation. α-actinin recruitment directly correlates with force generation within mature adhesion sites. FRET-based α-actinin sensors support that the protein is under tension, and that an increase in this tension at FAs happens alongside adhesion elongation and growth [81,82]. Further support for the mechanosensitivity of α-actinin is the irregularity in cellular protrusion–retraction cycles upon knockdown of the protein, suggesting a role in maintaining functional ECM rigidity sensing [83]. In terms of p130Cas, mechanical stimuli can trigger tyrosine phosphorylation of the protein in cells undergoing stretch. In this manner, force transduction primes p130Cas for phosphorylation, which can activate the small GTPase, Rap1, and initiate its activity in a number of important signaling cascades such as integrin signaling [84]. Interaction with vinculin via the SH3 domain of p130Cas has been suggested to regulate its mechanosensing function, as stretch-dependent phosphorylation is attenuated in cells lacking the p130Cas–vinculin interaction. Notably, disrupting the p130Cas–vinculin interaction also results in smaller FAs. This could be accounted for mechanistically by the p130Cas–vinculin interaction stabilizing the open, active conformation of vinculin [85].

This aspect of FA regulation is interesting to think about in the context of pathogenesis, wherein actin remodeling and altered actin dynamics likely impact intracellular tension states and tension-responsive pathways operating within the cell. For example, the obligate intracellular bacterium *Chlamydia trachomatis*, a causative agent of bacterial sexually transmitted disease, has been shown to modulate RhoA-dependent actin recruitment and myosin II activity to assemble an actin cage around its replicative niche, a membrane-bound vacuole called the inclusion [4]. Many intracellular bacteria restructure actin filaments and myosin motor proteins in order to form cage-like structures that protect the bacteria-containing vacuole within the cell. How might regulation of actin and the actin-cross-linking protein myosin II impact FA dynamics, which rely upon a stress fiber template and are known to respond to changes in mechanical force? Furthermore, intracellular pathogens such as *Rickettsia rickettsii* and *Shigella flexneri* are known to target the FA protein vinculin in order to disrupt cellular tension and promote intercellular spread [86]. This raises some intriguing questions—how does actin remodeling influence FA dynamics during different phases of a pathogen’s lifecycle, and are there other examples in which directly targeting FAs is a primary mechanism utilized by a pathogen to manipulate cellular tension? Additionally, are there microbial effector proteins or virulence factors which are themselves mechanosensitive and how might this facilitate host cell remodeling? Further investigation into the interplay between cellular tension and FA targeting during pathogenesis could be illuminating.

### 2.3. Calpain and Caspase-Mediated Cleavage of FA Proteins

Another method utilized by the cell to modulate FA dynamics involves the targeted proteolysis of select FA components. Specifically, the calpain family of cysteine proteases are capable of cleaving proteins to facilitate disassembly of adhesion complexes. Calpains are subject to tight regulation of their proteolytic activity. The best-characterized mechanism involves calcium activation. Indeed, the two major calpain isoforms, calpain-1 and calpain-2, can be differentiated based on a micromolar or millimolar requirement for calcium, respectively. An endogenous calpain inhibitor calpastatin also regulate calpain’s proteolytic activity [87]. Calpain-2 has emerged as the primary player in the regulation of FA disassembly, as knockdown of Calpain-1 has been shown to have little effect on proteolysis of FA proteins [88]. Initially, the link between calpain activity and integrin-mediated adhesion was characterized in the context of cell migration. It was found that inhibiting calpains reduced the ability of cells to migrate, as a consequence of large and stabilized adhesion complexes. This stability ultimately impaired cell detachment at the rear of the cell, thereby decreasing migration. The implication that calpains must therefore play a role in FA disassembly, spurred the search to identify which proteins calpains help to degrade. Known calpain targets include talin, vinculin, paxillin and FAK as well as α-actinin and p130Cas [89,90,91,92,93,94]. Though an exact consensus sequence predictive of calpain cleavage has not been elucidated, a multitude of computational approaches have been applied to develop calpain cleavage site predictive tools [95]. Interestingly, while capable of causing significant degradation, calpain often cleaves FA substrates such that the protein fragments retain a stable function, independent of the intact protein. Thus, calpain cleavage represents a permanent form of post-translational modification regulating FA dynamics.

Full-length talin consists of an N-terminal globular head domain as well as a C-terminal rod domain. Talin is one example of a FA protein whose protein fragments retain a functional role. For instance, it has been documented that once cleaved from full-length talin by calpain, the talin head domain fragment has a greater binding affinity for β3 integrin than the full-length protein [96]. In addition, overexpression of talin’s head domain enhances integrin activation and clustering [97]. The talin head domain has also been shown to exhibit greater affinity for the E3 ubiquitin ligase Smurf1 following cleavage, which marks the head domain for degradation [98]. This provides a mechanistic link between calpain-mediated cleavage and the downstream initiation of FA disassembly. While calpain can release the talin head domain from the rod [99], it also cleaves talin at two additional sites. One of these sites is located before talin’s dimerization domain [89]. Interestingly, the final cleavage site is only exposed upon a force-induced change to talin’s conformation. Post-translational arginylation appears to stabilize the half-life of this fragment [100]. A non-cleavable talin mutant (L432G) has been invaluable in parsing out how calpain cleavage and the liberation of talin head and rod fragments regulates FA dynamics. Classically, proteolysis of talin has been suggested to play a role in the turnover of mature adhesions. Recently, however, a role for talin cleavage in the early formation of adhesions was identified. Expression of non-cleavable talin impaired adhesion development, a defect which was rescued by the talin rod fragment but not the head fragment [101].

Interestingly, proteolytic processing of FA proteins has also been shown to play a role during pathogenic invasion. The facultative intracellular microbe *Bartonella henselae*, the causative agent of cat-scratch disease, forms an “invasome” scaffold at the site of entry which is comprised of FA proteins and promotes bacterial uptake. An RNAi screen performed in HeLa cells revealed that Src, FAK, β1 integrin, and the adaptor proteins paxillin, talin1 and vinculin are all essential components for invasome formation. Further investigation was then performed to validate the hits from the screen. A role for FAK and Src activity was implicated, as addition of the Src inhibitor SU6656 or expression of FAK with mutated phosphorylation sites (Y397F, Y861F) decreased invasome formation. Immunofluorescence confirmed that FAK pY397 and Src pY418 localize at F-actin ends, along with paxillin pY118, at the invasion site. Additionally, further siRNA knockdown experiments revealed decreased invasome formation upon knockdown of β1 integrin and talin1. It was found that specifically the extended active conformation of β1 integrin was required for efficient invasome formation. Deployment of the Type IV secretion system was reliant upon β1 integrin interaction during invasion, but not downstream FA signaling factors. Given that talin binds integrins via its FERM domain and facilitates their activation, the reliance upon both β1 integrin and talin is notable. This suggests that *B. henselae* utilizes the dual strategy of “outside–in” signaling through interaction with β1 integrin and “inside–out” signaling through talin-dependent activation of β1 integrins during invasome formation. To further parse out the mechanism of talin’s involvement, domain studies were undertaken. The dimerization and actin-binding domains of talin were not required for invasome formation, but liberation of talin’s head domain via calpain-dependent cleavage of talin was required. This indicates talin processing and activation of β1 integrin by the FERM domain may be important for building the invasome scaffold at the entry site of *B. henselae* [102].

The calpain family of proteases is involved in the regulated proteolytic cleavage of FA proteins in order to facilitate dynamic FA disassembly, a requirement for cell migration. Another cellular process in which degradation of adhesion components proves relevant is apoptosis. Additional proteolytic enzymes such as caspases have also been implicated in this process. FA proteins such as FAK, p130Cas and paxillin all engage in apoptosis suppression through pro-survival signaling. Specifically, FAK protects against apoptosis via stimulation of a number of signaling pathways including via activation of PI3K/Akt signaling, Ras GTPase signaling, anti-apoptotic NF-κB signaling as well as suppression of p53 expression levels [103]. FAK is known to interact with the adaptor proteins p130Cas and paxillin at FAs. Within fibroblast cells, p130Cas has a role in suppressing anoikis (apoptosis in response to detachment from the ECM), as evidenced by increased cell death in cells expressing a dominant-negative p130Cas-SH3 mutant. FAK interaction with paxillin also plays a role, since deletion of paxillin or SH2-domain binding sites abolished anoikis suppression [104]. Due to the ablation of adhesion-dependent survival signaling, the degradation of these adhesion components by calpain or caspase enzymes within anchorage-dependent cells can induce cell detachment from the ECM and cell death. FAK is cleaved by caspases during apoptotic cell death, whereas the degradation of paxillin and p130Cas has been shown to be context dependent and influenced by both the caspase and calpain families of proteases [94,105]. Apoptosis may be triggered by an array of stimuli deleterious to the cell, including reactive oxygen species (ROS), DNA-damage, or microbial infection.

Even though apoptosis induction functions as a host response, there are two sides of the coin when it comes to programmed cell death and host–pathogen interaction. Apoptosis is part of the innate immune response meant to eliminate pathogens before they cause productive infection. However, in some circumstances, apoptosis induction actually proves beneficial to the pathogen. Pathogens use a variety of complex mechanisms to regulate cell death, often suppressing cell death at certain stages of their replicative lifecycle while actively promoting it at others, which would permit their eventual dissemination. Viruses are known to induce apoptosis to ensure the dissemination of progeny virions. There is also evidence that promoting cell death is an advantageous mechanism for bacteria to spread to neighboring cells, evade or kill immune cells like macrophages, and gain the nutrients necessary for survival [106]. Enteropathogenic *Escherichia coli* (EPEC) is a Gram-negative bacterial pathogen, and a causative agent of epidemic diarrhea. It has been demonstrated that enteric pathogens such as EPEC can induce apoptosis during infection in host intestinal epithelial cells. Importantly, apoptotic cell death may be a contributing factor to the damage induced by infection to a patient’s intestinal mucosa, indicating a critical role for bacteria-induced apoptosis in the capacity of these bacteria to cause disease. Significantly, one of the main EPEC effectors involved in promoting epithelial cell cytotoxicity (EspC), does so by targeting FA proteins. By 3 h post-infection, EspC secretion induces cell rounding and cytotoxicity. This is dependent on the internalization of the effector and its functional serine protease motif. The endpoint of cell rounding was found to be cell detachment after 3 h of EPEC infection. Interestingly, introduction of EspC into a rabbit EPEC strain initially lacking the effector induced similar detachment kinetics. Cell detachment was a direct result of the cleavage of FA proteins, including fodrin, paxillin, and FAK both in vitro and in vivo [107]. Further modification to FA proteins included FAK dephosphorylation. FAK was also found to be more susceptible to degradation by the serine protease motif of EspC than paxillin. Since endogenous caspases and calpains also cleave FA proteins during cell death processes, it is interesting to note that EspC is associated with increased activity of caspases 3, 8 and 9. EspC-mediated cell death was found to proceed through an intrinsic or mitochondrial apoptosis pathway. EPEC infection of Hep2 cells caused an EspC-dependent increase in the translocation of the pro-apoptotic protein Bax, cytochrome C release from the mitochondria, as well as caused a loss of mitochondrial potential. While EspC protease activity is necessary for EPEC to induce cytotoxicity, both apoptosis and caspase cleavage could still occur in an EspC protease-dead mutant strain [108].

Another example of a pathogenic effector known to modulate FA dynamics in its pursuit to induce cell death is (E4orf4), the polypeptide encoded by the E4 open reading frame 4 of adenoviruses. Adenoviruses infect mucous membranes of humans, and are the causative agent of a range of common-cold or flu-like symptoms. The highly toxic nature of E4orf4 has led to speculation that E4orf4-induced cell killing may facilitate release of adenovirus viral progeny. The E4orf4 death pathway was characterized to be caspase independent. Instead, a FA-dependent pathway of E4orf4 cell killing has been proposed which involves E4orf4 interaction with Src, and dysregulation of Src signaling pathways. E4 co-precipitates with v-Src and c-Src and the E4orf4–Src interaction plays a functional role in E4orf4-induced cell death, as treatment with a selective Src kinase inhibitor PP2 inhibits the membrane blebbing normally induced by E4orf4 overexpression. Additionally, E4orf4 has been observed to modulate the kinase activity of c-Src, as the tyrosine phosphorylation levels of certain Src substrates were altered during overexpression. Indeed, E4orf4 causes increased blebbing and cell death in c-Src overexpressing or constitutively active mutants, but not kinase-dead mutants. Overall, by altering Src dynamics at FAs, E4orf4 causes the improper assembly of FAs, thereby disrupting pro-survival signaling and initiating cell death [109,110]. Further characterization of this novel death pathway revealed a contribution for additional cytoskeletal targets. Specifically, formation of a juxtanuclear actin–myosin network seems to drive the blebbing phenotype. This occurs alongside phosphorylation of the myosin light chain and subsequent activation of myosin II. This component of the pathway is also Src dependent, as cells expressing a mutant defective in Src binding did not exhibit phosphorylated myosin light chain (p-MLC). Treatment with the myosin II-specific inhibitor blebbistatin triggered the disassembly of this juxtanuclear actin ring structure and decreased the cytoplasmic pool of E4orf4. Actin manipulation is therefore an important factor driving E4orf4-mediated cell death, also as evidenced by a reduction in nuclear condensation during inhibition of Rho GTPases, myosin II or Arp2/3-dependent actin polymerization [111]. In addition to Src, changes in paxillin adhesion dynamics have also been implicated in the E4orf4 death pathway. The observation was made that E4orf4–Src signaling induced the activation of Jun N-terminal kinase (JNK). Providing further support for JNK’s role in the death pathway, the juxtanuclear actin network was reduced in JNK-depleted cells. Normally, JNK signaling is mediated by paxillin in the context of cell migration. This prompted a more in-depth look at paxillin in the novel context of JNK’s pro-death function. It was found that E4orf4-transfected cells had enlarged FAs, which could be decreased by JNK siRNA knockdown. Specifically, JNK-dependent phosphorylation of paxillin’s Ser178 site was identified as the driving force behind the reduced FA turnover and resulting stabilization observed during E4orf4 cell killing. The JNK pathway was shown to involve Src, Rho, and Rho kinase (ROCK). Accordingly, the downstream stability of FAs was reduced by ROCK and JNK inhibitors [112]. As it stands, the E4orf4 death pathway appears to hijack a Src–Rho–ROCK pathway, leading to a model where JNK-mediated phosphorylation of paxillin is a critical event facilitating the changes in adhesion dynamics and cellular tension which ultimately lead to cell death.

### 2.4. Autoinhibitory Mechanisms of FA Proteins

We refer the reader to a recently published and thorough review article on this topic [113]. Briefly, autoinhibition refers to the intramolecular interactions which may occur between separate domains within the same protein, and function to keep the protein locked in an inactive state. A variety of mechanisms exist which can relieve the autoinhibitory state, thereby liberating the protein and promoting its activation. This conformational change can be mediated by diverse cues such as post-translational modification, altered tension states, or proteolytic cleavage. The number of individual FA proteins which are governed by autoinhibitory mechanisms is significant, because it provides the cell with an additional point of regulatory control in modulating FA activity. Proteins can be maintained in an autoinhibited state until the appropriate context for their activation is met. When this regulation is compromised, such as in studies utilizing constitutively active mutants, altered or impaired FA dynamics are often observed. Properly controlled protein activation is therefore an important prerequisite for downstream FA signaling.

The adaptor protein talin adopts an autoinhibited conformation facilitated by interaction between the F3 region within talin’s FERM head domain and the R9 region of talin’s rod domain [114,115,116]. This autoinhibition has demonstrated functional consequences, as the interaction occludes the integrin and actin-binding sites within the talin molecule. A constitutively active talin mutant (E1770A), in which the F3–R9 interaction is disrupted, results in stable, mature FA complexes [117]. Vinculin’s autoinhibitory regulation is similar to that of talin, in that it also involves interaction between the head and tail domain (Vd1 and Vt; Vd4 and Vt) and these interactions are sufficient to block normally available ligand binding sites [118,119]. Several constitutively active vinculin mutants have been generated such as the (vinculin-T12) mutant demonstrated to reduce FA turnover [120] as well as the (T12-A974K) mutant meant to further destabilize vinculin head–tail interaction [121]. Once unfolding from an autoinhibited conformation occurs, subsequent protein–protein interaction may inhibit refolding, such is the case for talin which is locked into its unfolded conformation via its binding to the vinculin head domain [122]. Autoinhibition is a tightly controlled mechanism, in which specific signals are required to relieve the autoinhibitory state as well as prevent its inopportune refolding. This is especially important during dynamic events such as the initial formation of a FA complex. Another important protein which interacts directly with talin is RIAM, which aids in talin localization to the plasma membrane, thereby helping facilitate talin’s subsequent interaction with integrins. RIAM autoinhibition occurs between a region near the amino terminus of RIAM now deemed the inhibitory region (IN) and the RA domain at its Rap1 binding site. Mutations at either (E60A) or (D63A) abolished this binding, and both mutants were found to enhance colocalization of RIAM and Rap1 at the plasma membrane [123]. Interestingly, RIAM phosphorylation by FAK at a Tyr45 residue within the inhibitory region was shown to release RIAM from its autoinhibitory state, suggesting that RIAM’s autoinhibition and downstream ability to recruit talin are at least partially regulated by FAK [123].

FAK is also subject to autoinhibitory regulation. FAK’s autoinhibited structure involves interaction between the F2 lobe within its N-terminal FERM domain and the C-lobe of its kinase domain. The resulting closed conformation blocks access to FAK’s catalytic cleft and prevents the phosphorylation of its activation loop [124]. Therefore, in its inactive autoinhibited conformation, the residue Y397 is non-phosphorylated. In the context of embryonic development, a non-phosphorylatable (Y397F) FAK mutant was found to be early embryonic lethal, whereas embryos with a phosphomimicking (Y397E) mutation exhibited a comparably longer lifespan [125]. The other prominent tyrosine kinase which phosphorylates FA proteins, Src, can also exist in an autoinhibited state. Its inactive form is maintained by interaction between Src’s SH2 domain and a phosphorylated Tyr527 residue at the C-terminus of the protein. As a result, constitutive activation can be achieved through dephosphorylation or mutation (Y527F) at this residue [126]. The protein zyxin is also included in the list of FA proteins known to be autoinhibited by a head–tail interaction. Zyxin’s proline-rich “ActA” repeat region binds its LIM region to maintain an autoinhibitory state. A phosphorylation event at zyxin’s Ser142 residue was demonstrated to cause their dissociation, and a phosphomimetic mutant (S142D) alters cell behavior by preventing cell–cell detachment [127]. Finally, α-actinin’s autoinhibitory interaction is mediated by its calmodulin-like domain (CaM-LD) binding to its neck-R1 region [128]. A (NEECK) mutant has been developed to investigate α-actinin’s open constitutively active state [129].

It is becoming clear that autoinhibition acts as another point of precise regulatory control. Mechanistically, this allows FA proteins to switch between their inactive state and open active conformation in response to specific signals. Autoinhibition may very well mediate the ability of FAs to localize to the proper place at the proper time. An emerging concept within FA biology is the idea of “pre-complexes”, or the joining of groups of proteins prior to the actual assembly of a force-bearing FA linked to integrins [130]. These interactions, which form prior to the introduction of force, may be dictated by autoinhibitory maintenance of inactive states. For example, there is support for association between talin, vinculin and paxillin which occurs prior to the formation of an integrin-containing nascent adhesion [131]. These pre-complexes appear to perform a functional role in adhesion genesis, especially considering that nascent adhesions that do not contain both talin and vinculin exhibit impaired maturation dynamics. There is evidence of talin molecules at adhesion sites which are not immediately mechanically engaged. This suggests that not all adhesion molecules are instantly placed under force once targeted to adhesions [132]. Autoinhibitory dynamics may facilitate distinct pools of adhesion molecules which exist in different states and may function differently within the adhesion. At a given instance, vinculin has been shown to exist within at least three possible states, all dependent upon its particular adhesion-interacting partners. Inactive vinculin can be recruited by paxillin, whereas the talin–vinculin interaction promotes its activation state and facilitates its movement within the adhesion architecture [133]. Application of mechanical force can alter these conformational states, as autoinhibitory domains are held apart as the protein is stretched and therefore kept in its active state [134]. This role for inactive protein states, association of “pre-complexes” and unique pools of adhesion molecules raises intriguing questions for FA biology function during microbial infection. Many bacteria engage or recruit FA proteins to the site of invasion. Are such interactions largely transient or might these associations and possible “pre-complexes” go on to alter downstream FA behavior such as assembly and maturation? Additionally, while many FA proteins are autoinhibited and binding sites for protein interaction might be occluded at any given time, microbial effectors which mimic FA binding domains do not face the same regulation, and unlike their host counterparts could be considered to be constitutively active. It is conceivable that a pathogenic effector capable of binding a FA protein could alter the affinity between its autoinhibitory domains, activating the protein through modulation of its autoinhibitory state.

### 2.5. Nuclear Translocation of FA Proteins

The nanoscale architecture, or the precise protein distribution which exists within a FA can influence adhesion dynamics by spatially dictating which protein–protein interactions occur and how frequently. This nanoscale protein organization has been mapped using the super-resolution imaging technique iPALM, and revealed the existence of distinct protein-specific strata within FAs. These included an integrin signalling layer, a force transduction layer and an actin-regulatory layer, each comprised of different FA proteins [135]. Therefore, an important aspect of FA spatial regulation is determined by the functional compartments that exist within the adhesion architecture itself. However, there is another facet to the spatial constraints placed on FA dynamics, in that there are also functional compartments within the cell. Indeed, several FA proteins have been found to change their subcellular compartmentalization, by shuttling between FA sites and the nucleus. Therefore, spatial regulation of FA proteins operates not just at the level of adhesion architecture, but also the ability of FA proteins to properly translocate between the cellular compartments where they exert their function.

Zyxin and paxillin are two such FA proteins that have been demonstrated to cycle between the cytoplasm and the nucleus [136,137]. Notably, both are LIM domain-containing proteins. Studies with Leptomycin B, an inhibitor of Crm1-mediated nuclear export [138], were critical in revealing the nuclear accumulation of zyxin and paxillin when nuclear export is blocked [139,140,141]. For paxillin, phosphorylation of its LD4 motif has also been proposed to function as a signal for nuclear export [142]. Both proteins contain leucine-rich nuclear export signals (NES), but lack a canonical nuclear import sequence (NIS). This suggests that they may enter the nucleus via interaction with other carrier proteins. Proposed partners to aid in paxillin’s nuclear translocation include C-AbI, PABP-1 and FAK [142,143]. General adhesion dynamics, such as overall adhesion stability, have also been shown to influence the nuclear transport of paxillin. Signals which strengthened adhesions, such as a triangular micropattern or overexpression of FAK’s focal adhesion targeting (FAT) domain, reduced paxillin transport to the nucleus [144]. For zyxin, it has recently been suggested that its nuclear translocation is influenced by proteolytic processing. A zyxin fragment generated by the serine protease HrtA1 was found to translocate to the nucleus and protect from cell death [145]. The nuclear export inhibitor Leptomycin B has also been utilized to demonstrate FAK nuclear accumulation [146]. FAK contains both a nuclear localization signal (NLS) within its FERM F2 lobe and a nuclear export sequence (NES) located within the kinase domain. Additionally, the tyrosine kinase Src has a described nuclear function. Src uses a non-canonical, myristoylation-dependent pathway for nuclear translocation. A high content of nuclear Src correlates with low myristoylation status. Additionally, Src’s subcellular localization may be dictated by a myristoyl-binding site within its SH3 domain [18,147,148].

The biological functions that FA proteins exert once they translocate to the nucleus are a pressing question. Research has focused on elucidating what degree of transcriptional control nuclear FA proteins possess. Paxillin has been implicated in the regulation of transcription as well as mRNA trafficking. Specifically, paxillin association with the mRNA-binding protein, poly(A)-binding protein 1 (PABP1), has been shown to facilitate PABP1 nuclear accumulation [141]. Given PABP1′s reliance on paxillin for nuclear shuttling, and the important role it plays in the export of mature mRNAs to the cytoplasm, it has been suggested that paxillin is involved in the targeting of PABP1–mRNA complexes [141]. Additionally, interaction between paxillin and embryonic polyadenylation binding protein (ePABP) was shown to modulate androgen steroid signaling in a prostate cancer cell as well as frog oocyte (*Xenopus laevis*) model system. Paxillin’s role in androgen-mediated gene transcription in the oocyte model is supported by the observation that upon androgen stimulation, a paxillin–ePABP complex can enhance the translation of Mos mRNA, which leads to downstream oocyte maturation [149]. Additionally, paxillin appears to function as a nuclear receptor coactivator, as evidenced by its association with the androgen receptor (AR) and glucocorticoid receptor (GR) [150,151]. Finally, nuclear paxillin has been proposed to act as a transcriptional regulator of the *IGF2* and *H19* gene cluster, which provides a mechanism for nuclear paxillin’s regulation of cell proliferation. Paxillin modulates interaction between the enhancer region and promoter of each gene, to different effect. Promotion of this interaction activates IGF2 gene transcription whereas suppression of this interaction within the H19 gene downregulates H19 gene expression [152]. Zyxin nucleocytoplasmic shuttling has also been implicated in controlling gene expression. Notably, zyxin’s influence on gene transcription has been characterized in cellular tissues such as bone and smooth muscle which are responsive to mechanical stress. Zyxin has been shown to interact directly with transcription factors such as nuclear matrix protein 4 (NMP4), and may also indirectly enable interaction between NMP4 and p130Cas [153,154]. In renal epithelial cells, nuclear zyxin has been observed to stimulate the transcriptional activity of HNF-1β, an important regulator of cell differentiation [155]. Nuclear FAK can influence cell survival, through its formation of a p53 and E3 ligase mdm-2 degradation complex. By reducing levels of p53 in the nucleus, nuclear FAK contributes to enhanced cell survival [156]. Given its role as a tyrosine kinase, it is not surprising that one of the assigned functions of nuclear Src is to regulate the phosphorylation of other nuclear proteins [157]. There is also evidence that Src may influence chromatin structural changes, making chromatin more available to bind transcriptional factors [158].

Proper localization of FA proteins to adhesion sites or the nucleus is a fundamental checkpoint for their ability to exert their intended function. Unsurprisingly, processes that alter the translocation of FA proteins, whether through protein sequestration or by subverting the signals intended to regulate their translocation, can have drastic effects on FA and cellular behavior. Intriguingly, human papillomavirus (HPV) has been demonstrated to induce the nuclear accumulation of zyxin during infection, through the action of its E6 protein. E6 interaction with zyxin was identified from a yeast two-hybrid screen, and subsequent coimmunoprecipitation studies validated interaction both in vitro and in vivo. Zyxin’s LIM3 domain was determined to be essential for this interaction to occur. Some strains of HPV such as HPV-6 are considered low-risk strains, as they rarely develop into cancer. Interestingly, interaction between E6 and zyxin was selective for the low-risk HPV-6 strain, whereas the E6 protein of high-risk strains such as HPV-16, HPV-18, or HPV-11 did not interact with zyxin. Importantly, the striking nuclear accumulation of zyxin during infection had the downstream consequence of enabling zyxin’s function as a transcriptional activator. Specifically, zyxin’s proline-rich region was shown to have transactivating function in both yeast and mammalian cells, and this function was increased during E6 overexpression [159].

While interaction between E6 and zyxin is characteristic of the low-risk strain of HPV, the modulation of paxillin has been associated with the oncogenic potential of cancer-associated strains such as HPV-16. Both E6 from bovine papillomavirus (BE6) and E6 from the high-risk HPV-16 strain interact with the LD motifs of paxillin during infection, as indicated by yeast two-hybrid and immunoprecipitation studies. Specifically, BE6 binds the LD1 motif within paxillin, which functionally blocks paxillin’s ability to interact with its normal binding partners including FAK and vinculin [160,161]. Since FA proteins engage in multiple protein–protein interactions, pathogenic effectors which occlude these interactions can alter adhesion dynamics. For bovine papillomavirus, this interaction appears to be critical for inducing anchorage-independent growth as well as the disassembly of actin stress fibers. Similarly, overexpression of HPV-16 E6 disrupts the actin cytoskeleton, and its interaction with paxillin has been demonstrated to enable cellular transformation [161,162]. A BE6 construct in which an LD motif was fused to the amino terminus of the E6 was generated to study the importance of this interaction to cellular transformation. As the construct did not bind paxillin or induce transformation in C127 cells, it was concluded that the presence of a charged leucine motif alone is insufficient for transformation to occur, but rather something specific about paxillin’s LD motif as a cellular target is required [163]. Extending these studies, it was found that while anchorage-independent transformation depends upon paxillin’s LD motifs, tyrosine phosphorylation of paxillin at Y31 or Y118 is dispensable. However, paxillin mutants lacking LIM domains 1–3 did not support BE6 transformation. This LIM domain region regulates paxillin localization to FAs as well as FAK phosphorylation, suggesting a role for these processes in effective transformation [164].

### 2.6. Phosphorylation Events on Tyrosine and Serine/Threonine Residues of FA Proteins

Many FA proteins are regulated by post-translational modifications. Phosphorylation is a well-defined mechanism that is responsible for regulating the signaling events that occur at FAs (Table 1). Early work in the field established that integrin engagement with ECM results in robust tyrosine phosphorylation of many different FA protein components. Along with this observation, the discovery of FAK helped emphasize early on in FA research that protein kinases play a key role in signal transduction at FAs. Concomitant with the importance of protein phosphorylation is a role for phosphatases in dephosphorylating FA proteins, often as a means to regulate cell motility by inducing their disassembly. Indeed, achieving the proper balance between phosphorylating and dephosphorylating FA proteins has emerged as a potent regulator of FA function. For this reason, there has been considerable effort in recent years to map the phosphorylation sites of key FA proteins and assign these events a biological function. Phosphorylation has been implicated in varied aspects of FA behavior, from controlling expression level, proper subcellular localization, autoinhibitory interactions, and overall FA turnover dynamics. In this sense, the functional biological outcome of phosphorylation is heavily integrated with other regulatory processes acting to control FA dynamics. For example, phosphorylation may be the signal that induces relief from an autoinhibitory conformation, or the signal that promotes calpain cleavage.

Emphasis has been placed on defining the FA phosphorylation sites which regulate turnover dynamics. For example, phosphorylation at paxillin’s major Y31 and Y118 sites has been shown to promote cell migration. Along with the observation that adhesion disassembly is slower following mutation at these sites (Y31F; Y118F), it can be concluded that these phosphorylation events are involved in adhesion turnover [197]. Other phosphorylation events, such as vinculin’s Tyr100 or Tyr1065, can promote the activation of the protein through enhancing specific protein–protein interactions—in this case, between vinculin and its talin and actin-binding partners [180,181,182]. Others may regulate calpain-mediated proteolytic processing of the FA protein, such as talin’s Thr114, Thr150 or Ser446 [176,177,178]. Additionally, phosphorylation events have been implicated in promoting the proper intracellular localization of the protein, as is the case with phosphorylation of Ser139, Ser437, or Ser639 on p130Cas [184]. There are also quite a few defined phosphorylation sites which help modulate relief from autoinhibitory configurations. These include RIAM’s Tyr45, zyxin’s Ser142 and FAK’s Tyr194 residues [123,127,188]. Clearly, phosphorylation is a critical mechanism governing FA behavior, often by initiating further downstream regulatory processes or signaling.

Given the breadth of signaling cascades that phosphorylation can initiate, it makes sense that pathogens would take advantage of this facet of FA regulation to facilitate host cell remodeling. Phosphoregulation of adhesion proteins is a common pathogenic strategy utilized to initiate downstream changes to actin structures. The Gram-negative spiral bacterium *Helicobacter pylori*, the causative agent of peptic ulcer disease and a significant risk factor for gastric cancer, relies upon the dephosphorylation of vinculin to reduce FA numbers and lamellipodia formation during infection. Moreover, these disruptions correlated with impaired wound healing of adenocarcinoma gastric epithelial cells (AGS), indicating that reduced phosphorylation of vinculin is a significant mechanism underlying the tissue damage caused during infection. These changes were reliant upon the translocation of CagA, one of several Type IV-secreted effectors encoded by the *Helicobacter pylori* cag (cytotoxin-associated genes) pathogenicity island. CagA is encoded by virulent *H. pylori* strains but is often missing in less virulent strains, providing further evidence for its role in pathogenesis [198]. Phosphorylated CagA can inhibit the catalytic activity of the Src family of protein tyrosine kinases (SFKs) and it is this inhibition which has been shown to directly prevent the downstream tyrosine phosphorylation of vinculin at its functionally important Tyr100 and Tyr1065 residues. Reduced phosphorylation at these residues during infection was observed to impair vinculin’s interaction with the p34Arc subunit of the Arp2/3 complex, which caused reduced lamellipodia formation. Since lamellipodia aid in wound healing and cell spreading processes, this provides a mechanistic link between reduced vinculin phosphorylation and *H. pylori*-dependent tissue damage [199]. CagA also reduces the level of focal adhesion kinase (FAK) tyrosine phosphorylation during infection [200].

## 3. Pathogenic Microbes Utilize “Outside–In” and “Inside–Out” Signaling during Host Remodeling

Adhesion complexes at the cell surface transmit signals from the extracellular environment through receptor–ligand interactions that can result in a change in actin cytoskeletal structure, such as increased filopodia or stress fiber formation, which thereby alters mechanical force across the cell. The outcomes of these signaling events can lead to changes at the cell surface that alter the kinetics of extracellular particle uptake as well as changes at the basolateral membrane that can result in altered adhesion and motility. Microbes have evolved to alter receptor interactions to improve colonization of extracellular pathogens or to increase uptake and enable growth of intracellular pathogens. When microbes specifically engage with ECM components and surface receptors, they can induce receptor coupling and activation that results in activation of Rho GTPases, subsequent kinase and or phosphatase recruitment, as well as recruitment of actin-scaffolding proteins that enable rapid changes in cytoskeletal architecture. Other microbes secrete or express proteins that directly alter intracellular FA signaling, with outcomes that either increase spread of the pathogen by inducing cellular detachment or increase cellular adhesion to ensure a stable replicative environment for intracellular growth. In this section of the review, we focus on microbial modulation of both “outside–in” and “inside–out” signaling. These strategies are summarized in Figure 4 and some well-described examples are illustrated in Table 2. While eukaryotic pathogens have also been shown to target host cell adhesion, we are only focusing on mechanisms used by bacterial and viral pathogens in this review. Only a few instances of FA modulation by fungal pathogens are known [201,202]; however, numerous examples of manipulation of host cell adhesion by protozoan parasites have been described and we refer readers to a review on the topic [203].

### 3.1. “Outside–In Signaling” upon Microbial Engagement with ECM or Integrin Receptors

Clearly, changes to the ECM, transduced through integrin receptors, can influence FA dynamics. This same ECM–integrin–FA signaling axis is also relevant in the context of infectious disease. Pathogens interact with the ECM in order to adhere to and infect tissues. In fact, intracellular invasion is a critical occurrence leading to the virulence of many bacterial species. Host cell invasion is reliant upon proper adherence by the pathogen to the host cell surface, as well as induction of the requisite actin remodeling to promote bacterial uptake. Integrin receptors function at the nexus of ECM proteins and intracellular FA host cell signaling. Given the linkage between FAs and actin stress fibers, integrin-initiated signaling events also have the capacity to trigger downstream actin cytoskeletal rearrangement. Therefore, it is unsurprising that a common theme among pathogens during invasion is integrin engagement. In addition to facilitating bacterial attachment, this engagement can trigger early signaling events, such as inducing the recruitment or phosphoregulation of integrin-associated FA proteins [251].

Engagement with integrin during cellular invasion can occur through the action of microbial virulence factors. Some of these virulence factors are microbial proteins expressed at the surface of the bacterium, whereas others are deployed through specialized secretion systems once the bacterium has made initial contact with a host cell. Surface-exposed virulence factors which facilitate adhesive interactions between host cell proteins and a bacterium are termed “adhesins”. This class of molecule is essential for bacterial virulence, with many bacteria producing multiple adhesins [252]. Direct engagement often involves high-affinity binding between adhesin proteins and β1 integrins, as is the case for the protein invasin produced by *Yersinia pestis*, *Yersinia enterocolitica* and *Yersinia pseudotuberculosis* [253,254]. Other examples of direct engagement include the Ipa proteins produced by *S. flexneri* which can interact directly with α5β1 integrin [229]. The CagL protein of *H. pylori* also engages with the α5β1 integrin host cell receptor [217].

Some pathogens favor indirect association, in which interaction with ECM components such as fibronectin facilitates adhesion. These pathogens produce fibronectin-binding proteins (FnBPs) to promote indirect binding via fibronectin. The *Staphylococcus aureus* proteins FnBP-A and FnBP-B were some of the first FnBPs described to indirectly interface with integrin receptors [210]. Other FnBPs include *Streptococcus pyogenes* SfbI/Protein F1 [208] as well as the well-characterized CadF and FlpA made by *Campylobacter jejuni* [204]. The gastrointestinal pathogen *C. jejuni* colonizes polarized intestinal epithelial cells, and this attachment requires fibronectin and CadF. Inhibition of actin polymerization (Cytochalasin D, Mycalolide B) or microtubule dynamics (Nocodazole) prevented the internalization of *C. jejuni* but not its binding to INT 407 intestinal cells. Combined inhibitor treatment did not further prevent internalization indicating that actin and microtubules are involved in the same uptake mechanism. *Yersinia* is an interesting example of a pathogen utilizing both direct and indirect mechanisms of integrin engagement, as it also produces the virulence factor YadA for integrin binding via the ECM [212]. The adhesin NadA from *Neisseria meningitidis* shares structural similarity to YadA and also mediates β1 integrin-dependent bacterial adherence [214]. An interesting component of adhesin research is emerging evidence that close physical contact between a bacterium and its host cell serves as a prerequisite for the efficient translocation of secreted virulence factors. For example, research into the injection of *Yersinia* outer proteins (Yops) by the bacterium’s type III secretion system revealed that the presence of either invasin or YadA adhesin protein is sufficient to facilitate functional effector translocation [255]. Likewise, when the *C. jejuni* adhesin protein FlpA is mutated, impaired delivery of the pathogen’s (Cia) effector proteins is observed [205]. These data suggest important interplay between bacterial adhesins and translocated effector proteins during host cell invasion.

Viruses also make use of integrin engagement to facilitate host cell internalization. Many viruses accomplish this via Arg-Gly-Asp (RGD) peptides on their surface which mimics the RGD motif in fibronectin that plays a role in integrin binding. For example, adenoviruses bind to integrin αV to promote internalization, Kaposi’s sarcoma-associated herpesvirus binds integrin αVβ3, coxsackie virus A9 binds integrin αVβ6, HIV binds α4β7 and Ebola virus has been suggested to bind α5β1 integrin [256].

“Outside–in” signaling can occur upon microbial engagement with fibronectin and integrin receptors on the cell surface to trigger the formation of cytoskeletal structures that increase adhesion and/or phagocytosis. Engagement with host cell integrin receptors, whether by direct or indirect means, has the capacity to initiate signaling events modulating the activity of host cell FA complexes. Notably, integrins are not constitutively active, but rather must undergo a structural change from a bent-closed conformation to an extended-open conformation in order to bind ligands. In response to bacterial invasion, integrin clustering and activation are often observed. Depending on the integrin complex bound by the microbe, integrin activation will recruit kinases and adaptor proteins to activate Rho GTPases. Engagement of integrins, such as αVβ3, often results in Cdc42 or Rac1 activation, which can induce rapid actin polymerization at the cell membrane to create bacteria-engulfing protrusions such as ruffles, lamellipodia, and filopodia. The dynamics and protein content of these complexes can also be influenced by additional host receptors engaged by the pathogen. Conversely, engagement and activation of αVβ1 and α5β1 integrins can result in talin recruitment which binds F-actin and can serve as a scaffold for nascent FA formation [10]. Under increased tension, guanine exchange factors (GEFs) of RhoA are recruited to the integrin–talin–actin scaffold to activate RhoA, which in turn activates multiple effectors (e.g., ROCK, mDia, PI(4)P5K) that can increase actin polymerization and actomyosin contractility [257]. Further increases in tension and stress fiber formation often coincide with increased FA formation and stability [258].

Integrin clustering can also induce the recruitment of FA signaling molecules, and has been associated with the tyrosine phosphorylation of proteins such as FAK and paxillin. Indeed, a common observation during bacterial invasion is an increase in the tyrosine phosphorylation of these proteins. For *C. jejuni*, paxillin phosphorylation is increased during invasion, as determined with an anti-phosphotyrosine antibody. Infection studies with a CadF mutant revealed an increase in phosphorylation, but only when a 20-fold increase in MOI was utilized, suggesting that CadF as well as additional factors contribute to this process [206]. *C. jejuni* also phosphorylates EGFR during entry, which is dependent on its ability to bind fibronectin, indicating β1 integrin involvement. Additionally, FAK and Src have been implicated in promoting *C. jejuni* internalization, as evidenced by selective inhibitor studies [259]. Immunoblotting with anti-phosphotyrosine antibody has also been used during *E. coli* invasion to demonstrate an increase in the phosphorylation of both FAK and paxillin [260]. *H. pylori* has been shown to enhance the phosphorylation of FAK at six distinct tyrosine residues (Y397, Y407, Y576, Y577, Y861 and Y925) as soon as 30 min post-infection. Maximal expression levels for each residue were achieved at variable times post-infection, indicating multiple roles for FAK throughout infection. Mutation of another *H. pylori* outer membrane protein OipA was associated with significantly reduced FAK phosphorylation, whereas mutants within the cag pathogenicity island (PAI) only reduced phosphorylation at Y407. Phosphorylation of paxillin at residues Y31 and Y118 was also observed during *H. pylori* invasion [261]. However, siRNA-mediated knock-down of FAK inhibited paxillin phosphorylation, supporting the idea that *H. pylori*-induced interaction of FAK with paxillin is crucial for paxillin activation. Interestingly, it was reported that paxillin phosphorylation is downregulated during late infection. Similar to FAK, paxillin Y118 phosphorylation was decreased in both OipA and cag PAI mutants, and Y31 phosphorylation decreased only in the OipA mutant [262].

#### 3.1.1. TSA56

The tick-associated and obligate intracellular bacterium *Orientia tsutsugamushi* requires entry into endothelial cells to replicate. *O. tsutsugamushi* associates with α5β1 integrin on HeLa cells by utilizing the adhesin TSA56 that contains a fibronectin-binding domain. Ectopic expression of TSA56 fibronectin-binding domain alone could outcompete fibronectin binding and block internalization [220]. Consistent with signaling following β1 integrin attachment, actin was enriched at *O. tsutsugamushi* attachment sites and was associated with membrane protrusions grasping the bacterium. The FA proteins FAK and Src are tyrosine phosphorylated early during *O. tsutsugamushi* invasion, and inhibition of their phosphorylation by Genistein prevented internalization. The role of FAK during invasion was confirmed by a lack of *O. tsutsugamushi* entry in FAK^−/−^ MEFs. Consistent with kinase activation, phosphorylated FAK, Talin, and Paxillin localize to *O. tsutsugamushi* entry sites, and RhoA activation but not Cdc42 or Rac1 was observed during invasion. The RhoA inhibitor C3 exoenzyme from *Clostridium botulinum* prevented bacterial internalization, confirming the importance of RhoA function during entry [220]. While RhoA activation can lead to downstream signaling that results in increased stress fiber formation and adhesion, these phenotypes were not investigated post-invasion. Instead, decreased FAs and stress fibers have only been observed late in infection (5 days) when *O. tsutsugamushi* infected endothelial cells underwent apoptosis (TUNEL stain) and detached, which is consistent with reduced levels of the anti-apoptosis protein Bcl-2 [263]. Since endothelial cells of the blood vessel wall are slow to turnover, this detachment process is likely actively induced by *O. tsutsugamushi* to spread the infection.

#### 3.1.2. Opc

The extracellular Gram-negative bacterial pathogen *N. meningitidis* can cause life-threatening meningitis when it passes through the blood–brain barrier. *N. meningitidis* promotes uptake by human brain microvascular endothelial cells (HBMECs) through interaction of the bacterial surface protein Opc with fibronectin, resulting in activation of α5β1 integrin on the cell surface [264]. *N. meningitidis* internalization was decreased by inhibition of tyrosine kinases (Genistein) and Src kinases (PP2), indicating a role for tyrosine phosphorylation in entry events. Src-phosphorylation was increased upon infection and was dependent on both fibronectin and Opc. An invasive *N. meningitidis* strain induced tyrosine phosphorylation 2–4 h post-infection (hpi) of 65 and 125 kDa proteins, corresponding to migration of Src and FAK, respectively. Src involvement in invasion was confirmed as overexpression of the c-Src inhibitor CSK prevented *N. meningitidis* invasion in HEK293T cells. Expression of inactivated Src (K297M) or infection of Src-null cells (SYF) prevented *N. meningitidis* invasion, which could be restored by wild-type Src complementation. *N. meningitidis* infection increased the incidence of tyrosine phosphorylated proteins at FAs and induced stress fiber formation [207]. A follow-up study by the same group also implicated FAK in *N. meningitidis* entry. FAK inhibitor (PF 573228) blocked invasion, but not adhesion. Consistently, FAK^−/−^ MEFS also did not support invasion. FAK-dependent tyrosine phosphorylation of an 80 kDa protein (cortactin) was observed during *N. meningitidis* infection, and Src was shown to be required for both FAK and cortactin phosphorylation. Since cortactin is a key actin filament-interacting protein that regulates cortical actin structures, its role in meningococcal invasion was further dissected. Ectopic expression of cortactin with point mutations in the Arp2/3-binding residue in the NTA domain (W22A) or the dynamin-interacting residue in the SH3 domain (W525K) domains of cortactin indicated that these interactions were required for meningococcal invasion [265]. Despite the activation of α5β1 integrin and subsequent activation of FAK and Src kinases observed in these studies, the GTPases involved in propagating this signaling was not investigated. Similarly, Lambotin et al. implicated Rac1 activation and Src-mediated phosphorylation of cortactin during meningococcal invasion of endothelial cells as a consequence of activation of the endothelial receptor Erbb2 by lipo-oligosaccharide [266]. Despite the “outside–in” activation and recruitment of key FA signaling proteins (β1 integrins, FAK, Src), the role of other FA structural proteins in entry (e.g., vinculin, talin, paxillin) in meningococcal entry remains unknown, and their interrogation will be important for determining if a nascent adhesion complex is formed during invasion.

### 3.2. “Inside–Out Signaling” in Which Secreted Microbial Factors Signal from within the Cell

“Inside–out” signaling can occur in response to secreted microbial factors that alter the function of these proteins from within the cell and result in changes to adhesion complexes at the cell surface. Post-invasion, microbes can further alter FA protein complexes to alter the stability of attachment of the infected cell. Examples include increasing adhesion to support slower growing pathogens as in the case of *C. trachomatis* or decreasing contact with the ECM to induce rounding up of cells and spreading for the enteric pathogens *Y. pseudotuberculosis* and *S. flexneri*. Viral pathogens such as Human Papilloma Virus-16 alter FA signaling to induce loss of cellular adhesion, “transforming” host cells to become capable of anchorage-independent growth, the defining quality of virally-induced metastasis. Bacteria that utilize specialized secretion systems to inject effector proteins upon engagement with the mammalian cell surface can directly alter stress fiber and FA signaling from inside the cell. Several secreted effectors have been shown to mimic domains of known FA structural proteins, kinases, or phosphatases which gives pathogens the ability to alter FA formation and dynamics.

#### 3.2.1. YopH

Following increased attachment to the host cell by invasin’s interaction with β1 integrin receptor, *Y. pseudotuberculosis* is able to maintain cell contact and establish an infectious foothold on the cell surface while preventing phagocytosis. Phagocytosis-resistance is achieved by secreting the type III effector YopH into the eukaryotic cytosol, which dephosphorylates tyrosine residues on key signaling and FA proteins [227,267,268,269,270]. YopH consists of an N-terminal substrate binding domain and a C-terminal tyrosine phosphatase domain. Ectopic expression of a YopH phosphatase inactive mutant (Y403A) in HeLa cells revealed tyrosine phosphorylated substrates to be p130Cas, FAK, and paxillin [222,223,224]. As expected, these phosphorylated FA proteins localized to FAs along with inactive YopH (Y403A). Expression of inactive YopH inhibits *Yersinia* uptake, indicating that native YopH dephosphorylates p130Cas and FAK and re-localizes them to the cytosol. This is consistent with the observation that YopH-dependent dephosphorylation of p130Cas and FAK results in their absence from peripheral FAs [224]. Similarly, YopH was responsible for FA disassembly in J774 macrophages and was found to dephosphorylate FYB, p130Cas, SKAP-HOM, and p55 [225,226]. Decreased adhesion is consistent with findings that overexpression of YopH can induce cell detachment, which is also dependent on interaction with p130Cas [271]. A mutation within the N-terminal substrate binding domain (Q11A) of YopH prevented p130Cas binding and this Q11A mutant was readily phagocytized compared to wild-type *Yersinia*, confirming a role for p130Cas in phagocytosis-resistance during infection. The phosphatase-inactive mutant (Y403A) was even less resistant to phagocytosis. Complete removal of the substrate binding domain of YopH reduced *Y. pseudotuberculosis* burden in an intraperitoneal model of mouse infection, indicating that dysregulation of FA localization is an essential virulence strategy [272]. Further exploration of the pathogenic effects and bacterial burden in the intraperitoneal model with varying YopH mutant strains will help elucidate whether YopH’s ability to prevent phagocytosis, increased cell detachment, or both is required for full virulence.

Interestingly, a FA-targeting domain was found in the central region of YopH, and was found to contribute to phagocytosis resistance in cell culture and virulence in a intraperitoneal mouse infection model [270]. These findings inspired the model that upon secretion, YopH localizes to FAs to dephosphorylate FA proteins, which results in the collapse of FAs and its associated stress fibers. This could potentially alter the cytoskeleton such that uptake of the pathogen is inhibited. In parallel, re-localization of FA proteins to the cytosol by YopH could also play a direct role in inhibiting “outside–in” signaling from the invasin–β1–integrin interaction to prevent actin recruitment at the *Yersinia* attachment site.

#### 3.2.2. Certhrax

The anthrax disease-causing strain G9241 of *Bacillus cereus* produces the ADP-ribosylating toxin Certhrax, which induces cell detachment and cytotoxicity in epithelial cells and macrophages [237,238]. The mechanism of Certhrax-induced toxicity involves modulation of FAs through re-localization of vinculin to the host cytosol. Certhrax-transfected cells underwent retraction and lacked FAs, which could be partially restored by overexpression of vinculin, but not paxillin. Immunoprecipitation of Certhrax from HeLa cells revealed an interaction with vinculin, and subsequent analysis by MS revealed that Certhrax ADP-ribosylates vinculin at Arg-433 [239]. The effect of this post-translational modification on the ability of vinculin to interact with FA proteins and unfold into its “active” FA-associating form remains to be determined.

#### 3.2.3. OspE

The invasive bacillary pathogen *Shigella* causes massive damage to the intestinal epithelium to reach the underlying resident macrophages that support bacterial replication. Secretion of the type III effectors OspE and OspE2 by *Shigella sonnei* increases cellular adhesion to maintain an infectious foothold and enable efficient bacterial spreading through the colonic epithelium [243]. OspE2 prevents cell rounding during infection and prevents apoptosis, while OspE localizes to the ends of stress fibers and colocalizes with FAK and talin at FAs during infection [243]. Localization of OspE to FAs requires integrin-linked kinase (ILK). The kinase activity of ILK was specifically required as ectopic expression of ILK kinase-dead mutants in ILK^−/−^ MEFs did not restore OspE localization to FAs. The consequence of OspE localization to FAs was increased FA size and decreased sensitivity to nocodazole-mediated FA disassembly. The expression of both OspE and ILK in ILK^−/−^ MEFS also prevented cell migration in a wound-healing scratch assay. Expression of OspE also reduced tyrosine phosphorylation of other ILK targets, FAK and paxillin. Interaction between OspE and ILK is essential for these OspE-directed phenotypes, given that an OspE (W68) mutant that is unable to bind ILK does not increase FA size or decrease FAK and Paxillin phosphorylation. Importantly, OspE is required for virulence in vivo since colonization of guinea pig large intestines was also dependent on OspE expression [244].

OspE1 and OspE2 have also been shown to interact with PDLIM proteins through their C-terminal PDZ domains. Interestingly, OspE1 localized with PDLIM7 at the boundary of FAs and stress fibers [273]. OspE1/2 double mutants had reduced bacterial spreading which could be recovered by complementation with OspE1, but not with the OspE1 truncation mutant lacking the PDZ domain [273]. It is still unknown whether the PDZ domains of OspE1 contribute to its localization to FAs where they can interact with ILK, or if these PDZ domains play another unknown role in FA biology, such as stabilizing stress fibers and/or increasing FA maturation. The latter possibility is intriguing since increased tension across the cell might favor conditions for *Shigella* intercellular spread.

### 3.3. Effectors of Attaching and Effacing E. coli

EPEC and EHEC *E. coli* strains drastically modify the cytoskeleton of polarized intestinal epithelial cells to initially colonize the intestinal tract, and induce cell detachment to further the spread of infectious particles. However, early infection events and replication require attached host cells, thus EPEC-secreted effectors must balance changes to signaling that affect adhesion and detachment depending on the stage of infection. Cell detachment during EPEC infection is dependent on FAK dephosphorylation and also on cleavage of FA proteins by EspC [108,274]. The cytoskeleton modifying effectors secreted by pathogenic *E. coli* contribute to invasion by inducing the formation of actin-rich pedestals that ensure complete engulfment of the bacterium, followed by depolymerization of this structure to ensure efficient uptake. These drastic changes in cytoskeletal rearrangement upon invasion result in a marked increase in actin stress fibers and rapid cytotoxicity in the host cell. The bacterial effectors involved in pedestal formation activate the Rho GTPases Cdc42 and Rac1 in order to recruit host proteins and induce rapid actin polymerization, while effectors that alter stress fiber formation and prevent premature cell detachment largely target RhoA activity. Several recent reviews comprehensively describe the abundance of cytoskeleton-modifying effectors of pathogenic *E. coli* [275,276]. Here we focus only on secreted effectors that modulate RhoA activity and downstream signaling to alter cell adhesion and apoptosis during *E. coli* infection.

#### 3.3.1. EspO1

EHEC homologues to OspE of *Shigella* were investigated for effects on FAs. Both EspO1 and EspO1-2 from EHEC are required to prevent cell rounding and FA disassembly during infection, while neither are sufficient to prevent cell rounding on their own. Similar to OspE of *Shigella*, EspO1 and EspO1-2 precipitate with ILK. EspO1-1 localizes to FAs, while EspO1-2 aggregates in the cytosol. EspO1-2 also precipitates with EspM1, a RhoA activating guanine-exchange factor (GEF). RhoA activity and cell rounding is increased during infection with the EspO1/O1-2 double mutant, indicating that EspO1-2 also inhibits EspM1 GEF activity. Similarly, inhibition of the RhoA effector ROCK prevents the cell rounding phenotype in the double mutant [245]. The functional consequence of EspO-1 localization to FAs has not yet been investigated. Since it is a homolog to *Shigella* OspE that reduced ILK kinase activity and associates with ILK, it would be worth investigating if EspO-1 expression similarly reduces ILK-dependent tyrosine phosphorylation of FA proteins.

#### 3.3.2. EspM

A study of potential RhoA regulators in multiple pathogenic *E. coli* strains was undertaken based on the presence of WxxE-like motifs, that are known to target Rho-GTPases [248]. EspM1 secretion by EPEC increases stress fibers at the site of bacterial adhesion, while EspM2 secretion increased stress fiber formation globally. Secretion of EspM3, from the closely related species *Citrobacter rodentium*, induces radial stress fibers at the adhesion site. These EspM3-induced stress fibers were not observed in RhoA (N17) dominant negative-expressing cells or in ROCK-inhibited cells. Activated RhoA (GTP bound) was enriched in lysates expressing WT EPEC compared to uninfected and the enrichment was even more apparent in EspM2- and EspM3-expressing EPEC. Phosphorylation of the RhoA downstream effector cofilin was also increased in EspM2 and EspM3-expressing cells. Since phosphorylation of cofilin inhibits its ability to bind and sever F-actin, EspM2 and EspM3 are able to increase stress fiber formation by activating RhoA-ROCK-LIMK signaling [248].

Similar to the EPEC homologues, EHEC EspM1 and EspM2 are required for stress fiber formation, and the guanine nucleotide exchange factor activity of EspM1 is necessary to activate RhoA and induce stress fiber formation in EspM1-transfected cells [277]. However, EspM2 has also been implicated in pedestal formation and increased cell bulging. ZO-1 distribution at tight junctions in MDCK cells are re-localized when EHEC EspM2 is overexpressed, and leads to decreased leakage through the junctions. β-integrin redistributes to the basal side of the cell during EspM2 overexpression as well. Taken together, these EspM-2-induced changes allow extrusion of MDCK cells by tight junction re-localization to the base of the cell in a RhoA-dependent manner [249].

#### 3.3.3. EspG

Similar to EspM effectors, deletion of both EspG and its paralog Orf3 resulted in decreased stress fiber formation during EPEC infection. Neither deletion alone has this defect, and transfection of either EspG-GFP or Orf3-GFP alone in Swiss 3T3 cells was sufficient to recover stress fiber formation, indicating redundant function. Additionally, microtubules are disrupted under EPEC pedestals during early infection, and transcomplementation of the EspG/Orf3 double mutant indicated that both EspG and Orf3 are required for this disruption. EspG and Orf3 bind tubulin and stimulate microtubule depolymerization in vitro. To test whether the EspG-induced stress fiber formation was related to release of a microtubule-associated guanine exchange factor GEF-H1 during microtubule destabilization, dominant-negative GEF-H1 was transfected into cells prior to infection. Restoration of stress fibers in the EPEC triple mutant (dTir/dOrf3/dEspG) by EspG complementation was prevented in cells pre-transfected with dominant-negative GEF-H1. Similarly, siRNA-knockdown of GEF-H1 prevented EspG-dependent stress fiber formation in the triple-mutant strain. The effect of EspG expression on GEF-H1 re-localization from the cytoskeletal fraction to the cytosolic fraction was similar to that observed with nocodazole disruption of microtubules. Stress fiber formation during EPEC infection could be prevented by transfection with RhoA dominant negative (N19). RhoA activation during infection was EspG dependent as determined by RhoA immunoprecipitation with Rhoetekin. Inhibition of ROCK with Y27632 also prevented EspG-dependent stress fiber formation [278]. Taken together, these studies point to EspG and Orf3 functioning to disrupt microtubules and release GEF-H1, in turn activating RhoA activation and increasing stress fiber formation.

### 3.4. Vinculin-Mimetic Effectors

A common strategy of pathogens to co-opt FA protein complexes and signaling is to alter the localization of vinculin and/or its binding partners. *Shigella*, *Rickettsia*, and *Chlamydia* similarly secrete effectors that contain vinculin-binding domains (VBDs). By mimicking VBDs found in the vinculin binding partners talin and α-actinin, bacterial effectors can alter vinculin localization and function.

#### 3.4.1. IpaA

In contrast to the *Shigella* effector OspE that increases stress fiber formation and cell adhesion to facilitate cell–cell spread, the type III secreted effector IpaA from *S. flexneri* contributes to internalization by destabilizing adhesion complexes and inducing actin depolymerization. IpaA is required for efficient internalization during *S. flexneri* infection, and this was shown to be dependent on vinculin. Vinculin^−/−^ ASML cells internalized 10-times fewer bacteria than vinculin^+/+^ ASML cells. Immunoprecipitation of GFP-IpaA revealed interaction with vinculin as early as 5 m post-infection. *Shigella* induced rapid formation of an actin coat by 5 m that had disappeared by 30 m post-invasion. However, IpaA-deficient *Shigella* recruited actin to a similar extent by 15 m and continued to remain associated with actin at 30 m post-invasion, indicating the importance of both actin polymerization and depolymerization in the internalization process [279]. Purified *Shigella dysenteriae* IpaA binds vinculin with high affinity, effectively preventing the formation of talin–vinculin complexes in vitro. Electron microscopy and F-actin sedimentation assays revealed that F-actin depolymerized in the presence of both IpaA and vinculin, but not vinculin or IpaA alone. These findings inspired a model of IpaA association with vinculin at stress fibers to induce actin depolymerization and enable uptake of *Shigella* [228]. In addition to causing F-actin depolymerization, microinjection of purified IpaA into HeLa cells resulted in fewer peripheral FAs and more central adhesions, which is consistent with cells undergoing cell rounding [228]. Expression of the N-terminus (1–500 aa) of IpaA prevented interaction of talin and β1 integrin, which is consistent with decreased β1 integrin activation in IpaA-expressing cells [232]. Taken together, IpaA rapidly depolymerizes actin stress fibers by preventing the F-actin stabilizing vinculin–talin interaction and decreases integrin-ECM contacts by preventing β1 integrin activation by talin.

Cell rounding in IpaA-expressing cells was shown to be dependent on actin contractility following RhoA activation [232]. IpaA expression increased RhoA activity as demonstrated by rhotekin immunoprecipitation (GTP-bound Rho interacts with rhotekin) and by increased phosphorylation of the RhoA effector myosin light chain (MLC), which directly increases acto-myosin contractility [232]. While RhoA activation by bacterial effectors often results in increased stress fiber formation and adhesion, the opposite is true for *S. flexneri* IpaA-induced RhoA activation. This is likely due to IpaA’s ability to disrupt the talin–vinculin and β1 integrin–talin interaction, which would prevent the formation of new FAs. In the context of apoptotic cells that have cleaved FA proteins by caspases, RhoA is activated and leads to myosin contractility that causes cell rounding [280]. Thus, a similar phenomenon could be occurring in IpaA-expressing cells, which in the absence of FA anchorage of the cytoskeleton to the ECM, RhoA activation results in cell rounding and detachment.

Multiple IpaA–vinculin complexes have been resolved by X-ray crystallography, revealing the unique vinculin binding potential of three IpaA vinculin-binding sites (VBS). The first crystallized structure to reveal how IpaA can mimic the talin–vinculin interaction was of the *S. flexneri* IpaA C-terminus containing both VBS1_(559–587)_ and VBS2_(559–587)_ with the N-terminal head domain of vinculin_(1–258)_. Gel filtration analysis confirmed that IpaA VBS1-VBS2 bound two vinculin molecules [281]. A follow-up crystallization study revealed different binding interactions for VBS1 and VBS2. For IpaA VBS1, a canonical Vh1:VBS structure was observed. VBS1 incorporation induced a conformational conversion of the Vh1 N-terminal 4-helical bundle (H1-4) to a 5-helical bundle (H1-5) [282]. Thus, IpaA VBS1 mimics the VBS sites of talin that displace the vinculin tail domain and activate vinculin’s actin-binding domain [230]. A unique structure of VBS2 interacting with vinculin α-helices 2–3 and with the Vh1 C-terminal 4-helical bundle, while the conformation of C-terminal bundle remained unchanged by interaction with VBS2. It was determined that the binding of VBS2 to the C-terminal Vh1 bundle mimics non-activating binding of talin-VBS3 to vinculin [282].

Surface plasmon resonance assays indicate that purified IpaA VBS1 interacts with the vinculin head (Vh) domain with a higher affinity than VBS2. Extremely high affinity (nanomolar) interactions were found between the IpaA C-terminal domain, containing VBS1 and VBS2, and full-length vinculin, with a very slow off rate [230]. Recruitment of vinculin to *Shigella* entry sites requires both IpaA VBS1 and VBS2, and results in an extensive actin-vinculin cup that surrounds the bacterium. Interestingly, VBS-deficient *Shigella* only recruited actin to the base of the bacterial contact site. The N-terminus and VBS1 of IpaA is sufficient for efficient internalization and dissemination, diminishing the role of VBS2 [230]. Both VBS1 and VBS2 can displace α-actinin from Vh, but cannot displace talin. Neither talin or α-actinin could displace IpaA VBS1-2, VBS1, or VBS2 from Vh, indicating that IpaA likely interacts with the free vinculin pool instead of that associated with actin complexes. However, interaction of IpaA VBS1-2 with full-length vinculin was sufficient to activate vinculin’s F-actin-binding domain [230]. Actin filament elongation from spectrin-actin seeds was blocked by vinculin alone, but the actin-capping activity of vinculin became leaky when in complex with the IpaA C-terminus [283].

Recent studies of a third centrally-located VBS3 of IpaA revealed its role in mediating interactions with talin during entry. The vinculin-binding capacity of VBS3 was confirmed by gel filtration, given that an extended C-terminus of IpaA complexed and eluted with three molecules of vinculin [281]. VBS3 of IpaA also binds Vh1 and contributes to efficient invasion by *Shigella*. Similar to VBS1, VBS3 can disrupt the Vh–Vt interaction and the association of vinculin with F-actin in vitro [231]. Actin association with *Shigella* during invasion is dependent upon talin, given that GFP-talin associates with *Shigella* upon entry, and internalization is reduced in talin-depleted cells. Interestingly, *Shigella* mutants missing IpaA VBS1-2 or VBS3 did not form talin coats. The N-terminus containing VBS3 bound the partially stretched Vh1-Vh4 domain of talin [247]. This interaction was confirmed as crystallization of VBS3 of IpaA with vinculin H1-H4 confirmed that VBS3 mimics the H5 domain of talin to interact with vinculin. Similarly, VBS3-GFP localized with talin and vinculin at FAs, and this localization was prevented by talin or vinculin depletion. In contrast to observations made in cells expressing full-length IpaA, the size and number of FAs was reduced in cells expressing GFP-VBS3 point mutants compared to wild-type GFP-VBS3, indicating a FA-stabilizing role for VBS3. A unique role for VBS3 of IpaA during *Shigella* attachment was observed. Lamellipodia and filopodia form at sites of *Shigella* attachment to induce uptake by the effector IpaC. Of note, *Shigella* infection of cells during replating and substrate stiffness assays revealed IpaA-VBS3, vinculin and talin labeling at the base of filopodia. It was observed that filopodial extensions were longer due to stabilization at the base in VBS3-labeled adhesions, increasing instances of filopodial capture of *Shigella* [247].

#### 3.4.2. Sca4

Intercellular spread by rickettsial pathogens depends on the ability to cross cell–cell junctions. Vinculin maintains tension and increases barrier function at cell–cell junctions by acting as a tether between the actomyosin network and α-catenin. *R. rickettsii* secretes surface cell antigen 4 (Sca4) to regulate the actin cytoskeleton by mimicking α-catenin’s interaction with vinculin. Two vinculin-binding sites on Sca4 are responsible for binding and activating vinculin. Unlike YopH and IpaA, transfection of Sca4 did not induce morphological changes in NIH-3T3 cells, but full-length Sca4, Sca4_VBS1(406–585)_ and Sca4_VBS2(772–1008)_ localized with vinculin at internal FA sites rather than at the leading edge [235]. The *Rickettsia parkerii* Sca4 ortholog was shown to be secreted upon invasion, but did not localize with the bacteria and instead localized to punctae in the host cell cytosol. HA-Vinculin (Vh) immunoprecipitated Sca4 when co-expressed in HEK293 cells, but did not precipitate a Sca4 mutant with multiple VBS point mutations (Sca4-VBS-NC). An *R. parkerii* Sca4::tn strain was not able to spread to neighboring cells in an infectious focus assay. Cell-to-cell spread was rescued by transcomplementation with Sca4-FLAG but not with Sca4-VBS-NC-FLAG. RNAi silencing of vinculin did not alter wild-type *R. parkerii* spread, but rescued spread by the Sca4::tn mutant, indicating a negative regulatory role of vinculin on *R. parkerii* spread that can be overcome by Sca4–vinculin binding. Competition between Sca4 and alpha-catenin for interaction with vinculin was demonstrated by immunoprecipitation of purified recombinant proteins. Sca4 outcompeted alpha-catenin interaction with vinculin, while VBS-mutated Sca4 did not. These findings inspired a model of Sca4-dependent inhibition of vinculin, such that vinculin could not increase cytoskeletal tension at cell–cell junctions. This was further supported by the observation that blebbistatin-induced stress fiber disassembly recovered intercellular spread of the Sca4::tn mutant [236].

#### 3.4.3. TarP

*Chlamydia* are obligate intracellular pathogens that rely on a biphasic developmental cycle to propagate and spread infection in mucosal epithelia. The invasive form of chlamydiae is a small dense non-replicative form that must be endocytosed in order to establish an intracellular niche. Once endocytosed, *Chlamydia* undergo primary differentiation into its metabolically active and replicative form. The role of the chlamydial translocated actin recruitment phosphoprotein (TarP) from multiple chlamydial species has been extensively studied to characterize its role in ensuring efficient invasion of epithelial cells, with a special focus on TarP’s multiple signaling and actin-binding domains [4,233,241,284,285,286,287,288]. *Chlamydia caviae* recruit vinculin at the site of invasion with three C-terminal vinculin-binding domains. TarP-VBS1 alone localizes with vinculin similar to the domain containing all three VBS. Interestingly, immunoprecipitation of vinculin was much more robust with full-length HA-TarP and the entire VBD (VBS1-3) than VBS1 alone, indicating additional roles for VBS2 and VBS3 in vinculin binding. The LD and VBD domains independently recruit pFAK and vinculin, respectively. While these domains enrich F-actin separately, transfection of the full LDVBD robustly recruits actin, pFAK (Y397) and vinculin [233].

Crystallization of the most C-terminal VBS1 of TarP with Vh revealed a strikingly similar formation of a 5-helical bundle similar to that formed between talin and vinculin. Gel filtration analysis of talin–vinculin complexes in the presence of TarP-VBS1 indicated displacement of talin from vinculin [289]. This is in contrast to findings that the LDVBD domain of *C. trachomatis* TarP has a stabilizing role on FAs as indicated by their resistance to blebbistatin [234]. However, the affinity of talin for vinculin is increased under mechanical force and thus may not be easily displaced by the VBS1 of TarP at FAs. Furthermore, the contribution of the individual VBS domains of *C. trachomatis* TarP to vinculin binding and FA formation/stability has not yet been investigated. The TarP homologues in *Chlamydia pneumoniae* and *C. trachomatis* serovar L2 also contain multiple vinculin-binding domains that are responsible for co-localization with vinculin at FAs [234,289].

Comparison of the crystal structures of the vinculin-binding domains from IpaA, Sca4, and TarP revealed that VBS-containing proteins are stretched through interaction with F-actin-associated proteins (talin, α-actinin, Riam, α-catenin). The Vh can bind cryptic VBS sites and change to an open formation to induce new interactions with the Vt. Atomic detail of force transmission at vinculin–VBS interactions during shear and zipper-like pulling geometries indicates stabilization and dissociation of the interaction, respectively. Protein-specific VBS sites, VBS orientation, and the direction of force transmission contributed to molecular changes observed at the vinculin–VBS interaction site [290].

*C. caviae* also recruits phospho-FAK immediately during invasion, and this recruitment can be mimicked by the membrane-bound paxillin-like LD domain present in the C-terminus of TarP. TarP-LD is sufficient to directly bind and recruit FAK. TarP-LD also recruits Arp2/3 and F-actin in a FAK- and cdc42-dependent manner. Inhibition of FAK recruitment (FAK^−/−^ MEFS), Arp2/3 activity (CI-666), or cdc42 activity (DN-Cdc42) prevented efficient invasion of *C. caviae*, implicating the LD domain of TarP in initiating signaling required for *C. caviae* invasion of epithelial cells. *C. caviae* rapidly recruits phospho-FAK to the site of entry, and FAK is required for efficient entry. The highly conserved LD motif of chlamydial TarP is sufficient to recruit FAK at membrane sites, similar to that of LD2 of human paxillin. The identified LD domain of *C. caviae* interacts with the FAT domain of FAK and colocalizes with FAK to FAs and the ends of stress fibers in LD-transfected cells. This Tarp-LD domain is sufficient to recruit Arp2/3, actin and cdc42, all of which are required for efficient invasion. Inhibition analyses revealed that Tarp-LD recruitment of cdc42 is dependent on FAK (FAK^−/−^ MEFs), and Arp2/3 (CI-666) and subsequent actin polymerization is dependent on cdc42 (DN). Taken together, the LD domain of *C. caviae* TarP orchestrates signaling through FAK-cdc42-Arp2/3 to induce actin polymerization at the invasion site, increasing invasion efficiency [241].

In addition to involvement in pseudoadhesion formation during chlamydial entry, the C-terminus of *C. trachomatis* serovar L2 TarP, containing the conserved LD and VBD1 domains, localizes to FAs and dysregulates FA turnover [234]. Depending on the species, *Chlamydia* require 2–4 days to maximize the number of chlamydiae that have undergone secondary differentiation into its infectious form, prior to their release outside the host cell. Since most mucosal epithelia are highly turned over, *Chlamydia* likely have evolved mechanisms to slow this turnover process long enough to complete development. Pedrosa et al. tested this hypothesis by studying the localization and FA-modulating activity of TarP [234]. *Chlamydia*-infected COS-7 cells displayed increased FA numbers and size, which is indicative of increased stability. Destabilization of stress fibers with blebbistatin triggers FA-disassembly in mock-infected cells, while *Chlamydia*-infected cells were resistant to blebbistatin-induced disassembly of FAs, indicating that infection stabilizes FAs. Localization of ectopically-expressed TarP to FAs in MEFs is dependent on vinculin, but not FAK. Indeed, resistance to blebbistatin-mediated FA disassembly was shown to be dependent on vinculin for both infected and TarP_LDVBD_-transfected cells. Mature FAs contain the protein zyxin, which is enriched in blebbistatin-resistant FAs, since they are naturally the most stable. *Chlamydia* infection results in primarily zyxin-labeled FAs, which are retained after blebbistatin treatment. Clues to how chlamydial TarP ensures stability of FAs was revealed by super-resolution microscopy by interferometric photoactivation and localization microscopy (iPalm), which is capable of capturing nanoscale molecular information about protein complexes near the base of the cell. iPalm imaging of *C. trachomatis*-infected epithelial cells showed that paxillin and FAK were repositioned from the FA signaling layer to the force-transduction layer at 24 hpi. Since repositioning of FAK and paxillin away from the signaling layer may prevent crucial interactions and events necessary for FA disassembly, these results indicate a possible mechanism for the observed increase in FA maturity and resistance to blebbistatin during infection. Ectopic expression of TarP also resulted in a repositioning of FAK and paxillin to the force-transduction layer of the FA. Since FA disassembly is required for cell motility, it is not surprising that *Chlamydia* infection inhibits both cell migration and detachment by trypsinization. Increased cell adhesion could be advantageous for *Chlamydia*, as they require 48–72 h to complete their biphasic developmental cycle. A post-invasion role for TarP in decreasing cell motility and detachment during infection might prevent premature exfoliation of the cell prior to the release of infectious *Chlamydia* [234].

### 3.5. Manipulation of FA by Viral Proteins

#### 3.5.1. KSHV TK

Kaposi’s sarcoma-associated herpesvirus (KSHV) Orf21 of KSHV encodes a thymidine kinase, TK, that has been recently discovered to also have tyrosine kinase activity that is required for lytic replication. In TK-expressing cells, tyrosine phosphorylated FA proteins (FAK, Paxillin) are re-localized from peripheral adhesions to internal sites. Cells ectopically expressing KSHV-TK contract and lose FAs, which results in membrane blebbing and cell detachment. Inhibition of RhoA signaling by multiple methods (DN N19 mutant, ROCK inhibitor, myosin II inhibitor) prevented cell contraction and increased stress fiber formation in TK-expressing cells. Similarly, TK-induced cell rounding could be blocked and paxillin localization at FAs could be restored with ROCK inhibitor. KSHV-TK autophosphorylates residues Y65, Y85 and Y120, which were determined to be required to induce FA disassembly and cell rounding. It was determined that the kinase activity of TK is necessary for its interaction with FAK. FAK’s central role in TK-mediated pathogenesis was confirmed by the observation that TK expression in FAK^−/−^ MEFs is not cytotoxic. Similarly, siRNA knockdown of paxillin also prevented TK-dependent FA disassembly and cell contraction. Thus, FA protein modulation is the cornerstone of TK-mediated cytotoxicity. Additionally, TK immunoprecipitated with multiple Crk proteins in a Y65/Y85-dependent manner, indicating a specific interaction with CRK SH2 domains. This was confirmed by the observation that overexpression of the SH2 CrkII or CrkL prevented cell rounding in TK-expressing cells. An additional interaction between TK and PI3Kp85 in a Y120-dependent manner was also observed, but its relevance to TK-mediated cytotoxicity remains unknown [242].

#### 3.5.2. Tat

Exposure of human brain microvascular endothelial cells (HBMECs) to exogenous HIV-1 transcriptional activator protein Tat was shown to increase FA formation when plated on laminin or fibronectin. Increased adhesions were dependent on signaling through VEGFR and on FAK activation. Expression of the C-terminus of FAK only (FRNK) blocked FA formation and migration during exposure of the cells to Tat. Within 5 m, Tat treatment increased FAK phosphorylation at Y20. HBMEC permeability was increased after Tat exposure, as indicated by the accumulation of Lucifer yellow in the bottom of a transwell coated with confluent cells [240]. This ability of secreted extracellular Tat to change the permeability of blood–brain barrier cells through FA modulation is similar to invasive mechanisms employed by *N. meningitidis*. Further study of possible “outside–in” signaling that may activate integrins to stimulate FAK phosphorylation during exposure to secreted Tat is needed to understand this invasive strategy.

#### 3.5.3. E7

In multiple HPV types, E7 binds the RhoA-inactivating hydrolase, p190RhoGAP, and dysregulates its Rho-activating function. An apparent decrease in RhoA activity was observed in E7-transfected cells as indicated by reduced F-actin and cell area. The p190 binding residues on E7 were detected and were found to be necessary for these changes [250]. The contribution of E7 modulation to pathogenesis during HPV infection and its relevance to cellular transformation still remains to be elucidated.

## 4. Conclusions and Summary

Pathogenic microbes co-opt host FA-regulatory mechanisms to ensure efficient uptake, survival, and dissemination. A common strategy utilized by these pathogens to manipulate FAs is to mimic the domains of host FA proteins, typified by the leucine–aspartic acid motifs (LD) and vinculin-binding domains (VBDs) present in several bacterial effectors. Given the shared domain structures of many FA proteins, we expect that many more unidentified FA modulators are present in human pathogens.

Studies that elucidate the role of microbial factors in the formation and/or disassembly of stress fibers and adhesion structures utilized well-characterized tools for studying FA biology (Table 3). These include genetic approaches such as truncation of FA proteins to eliminate key binding domains and mutation of key serine or tyrosine residues to alter the phosphorylation state and activation of FA proteins. Mouse embryonic fibroblasts deleted for FA genes have proven ideal for using complementation to dissect microbe-driven phenotypes. Additionally, chemical inhibitors that destabilize cytoskeletal components or target the activity of kinases have been useful for specifying the forces and pathways responsible for changes in FAs. Since the beginning of FA research, technical advances have rapidly helped further our understanding of adhesion dynamics. Mechanotransduction pathways have been elucidated using novel FRET-based tension sensors and traction force experiments. iPALM has facilitated more precise mapping of adhesion architecture. There is a greater understanding of the FA maturation process and the ultimate fate of distinct pools of adhesion molecules. These approaches are especially useful for large multi-domain proteins, such as TarP and Sca4, that could interact with multiple FA proteins and layers at the same time, and possibly differentially based on cellular tension. Some of these advances that have yet to be applied to microbial research could be useful new tools for dissecting microbe-driven phenotypes in greater detail and to further elucidate the molecular mechanisms of FA manipulation.

As outlined in this review, the large number of proteins involved in FA regulation and stability makes dissecting microbe-driven mechanisms in this process very complex. The functions of many of the described molecular factors (adenovirus e4orf4, HPV E7, HIV-1 Tat, *O. tsutsugamushi* Sca4) in this review have been detected in the context of ectopic overexpression. Now that these phenotypes have been described, the importance of these assigned functions in microbial infection should be better elucidated. One such approach would be to use well-described MEF deletion cell lines to prevent the expected FA-modulating phenotype during infection. In this context, restoration of the expected phenotype can be determined by ectopic expression of specific domains of the deleted FA protein targeted during infection, and further utilization of microbial strains that have been genetically modified to be defective in the FA-targeting microbial factor can be used to understand the complexity of these interactions during infection. Similarly use of double- and triple-mutant strains should be further utilized to determine the degree of cross-talk when multiple microbial effectors are targeting the same pathway (e.g., RhoA targeting by *E. coli* effectors, multiple adhesins interacting with β-integrins). For this reason, the approaches utilized and aspects of FA biology investigated in the reviewed work are highly varied. While the mechanisms of FA manipulation by a few effectors (IpaA and YopH) have been extensively studied, most of the microbial factors described in this review have only been described in a handful of publications, leaving many aspects of FA biology available for further exploration. Notably, changes to FA dynamics during infection have been tied to important mediators of pathogenesis during the course of disease, such as tissue damage. A more in-depth understanding of the mechanisms driving altered FA dynamics during infection offer exciting opportunities for therapeutic intervention. In fact, several FDA-approved kinase inhibitors have been shown to inhibit TK activity and lytic activation in vitro and to prevent KSHV-induction of tumors in mice [291]. Thus, targeting the ability of TK to modify FA proteins is a promising approach for preventing lytic reactivation of KSHV infection in AIDS patients. Similarly, we expect further study into FA manipulation by other pathogenic microbes to prove beneficial to the development of novel anti-microbials and ultimately help reduce the healthcare burden posed by infectious disease.

## Figures and Tables

**Figure 1 ijms-22-01358-f001:**
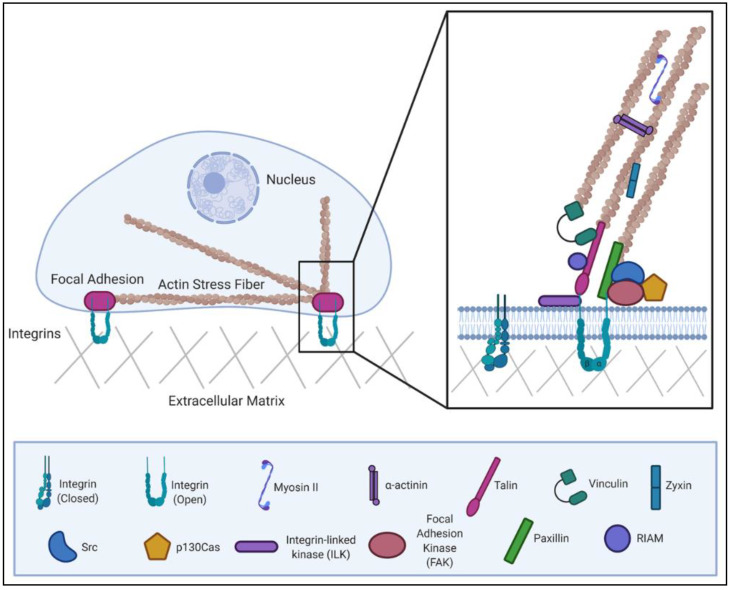
Diagram of a host cell focal adhesion complex. Focal adhesions link integrin receptors bound to extracellular matrix to intracellular actin within the cell. Inset highlights select protein components that comprise a focal adhesion and that are discussed within the review.

**Figure 2 ijms-22-01358-f002:**
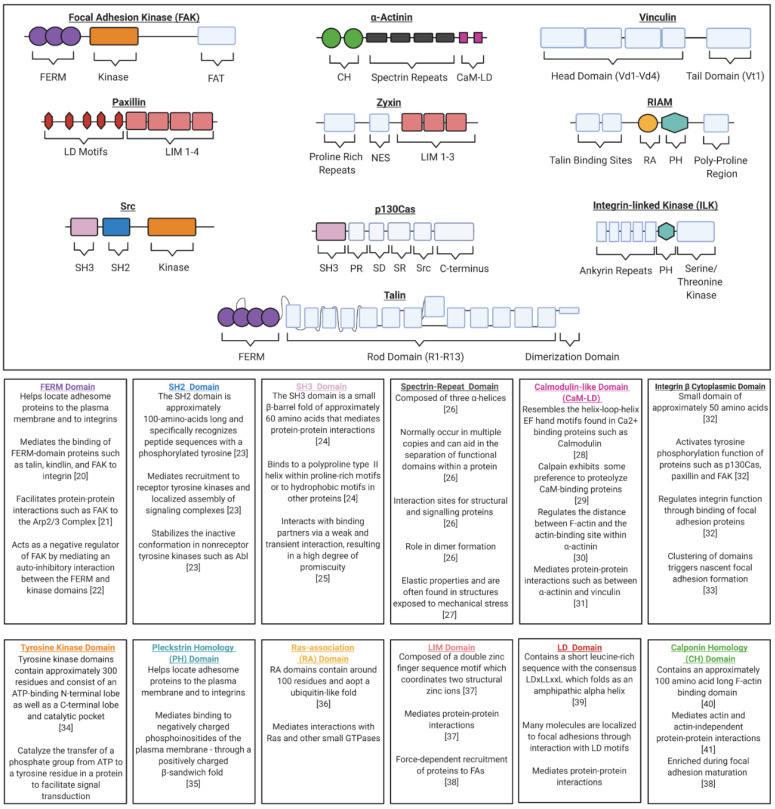
Diagram of structural domains for the focal adhesion proteins covered within the scope of this review. For p130Cas (PR = proline rich; SD = substrate domain; SR = serine rich). Additionally, the functional role for some of the most common protein domains relevant to focal adhesion biology are summarized below [20,21,22,23,24,25,26,27,28,29,30,31,32,33,34,35,36,37,38,39,40,41].

**Figure 3 ijms-22-01358-f003:**
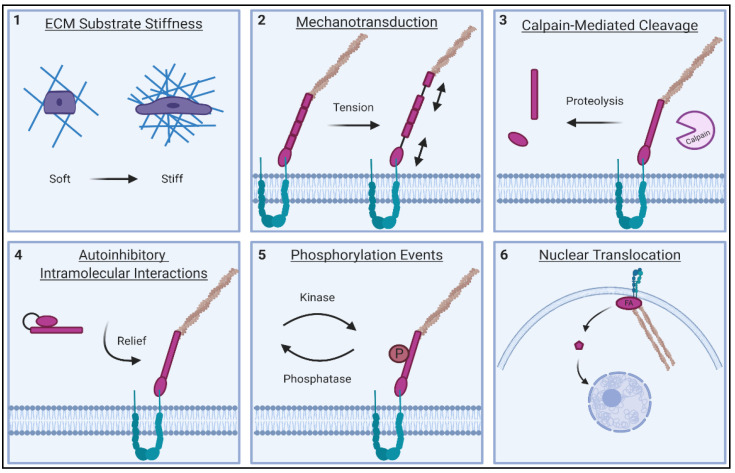
Simplified schematic depicting the FA-regulatory mechanisms discussed within this review. (**1**) ECM substrate stiffness: stiff substrate stimulates cell spreading and increases the average size of FAs. (**2**) Mechanotransduction: actomyosin contractility can promote protein unfolding and expose protein-binding sites. (**3**) Proteolytic cleavage: proteases such as the calpain family can cleave FA proteins. (**4**) Autoinhibitory interactions: protein activity can be regulated by intramolecular interactions wherein protein-binding sites are masked or exposed following relief of an autoinhibitory conformation. (**5**) Post-translational modifications: tyrosine phosphorylation is a key mediator of signal transduction at focal adhesions. (**6**) Nuclear translocation: several FA proteins have been shown to shuttle to the nucleus.

**Figure 4 ijms-22-01358-f004:**
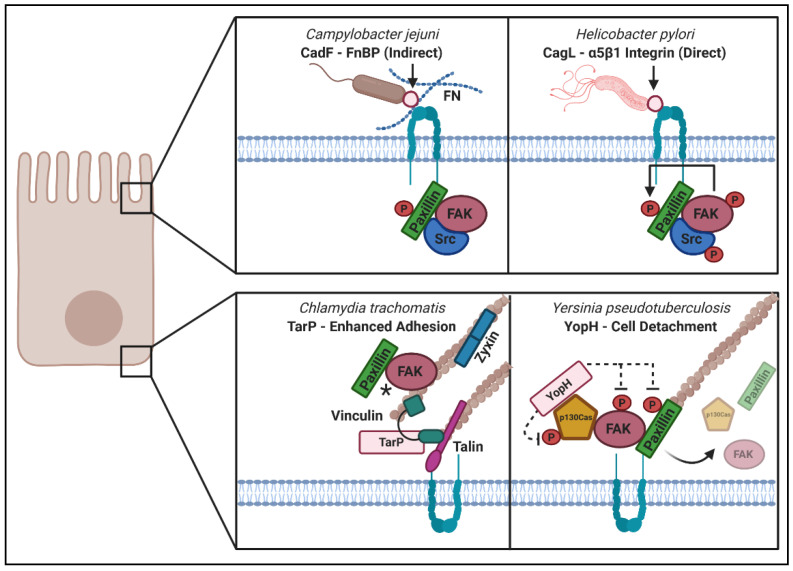
Schematic overview listing select characterized mechanisms by which pathogens modulate FA formation. During host cell invasion, a common strategy involves engagement with integrin host cell receptors, either directly or indirectly via fibronectin (FN) and microbial fibronectin-binding proteins (FnBPs). *Campylobacter jejuni* expresses the FnBP CadF which is necessary for cell invasion and has been shown to contribute to the phosphorylation of paxillin during invasion. The *Helicobacter pylori* effector CagL directly interacts with α5β1 integrin and induces the phosphorylation of Src and FAK, which in turn phosphorylates paxillin. Post-invasion, microbes have been demonstrated to modulate FA dynamics to promote maintenance of infection (enhanced adhesion) or in order to spread (cell detachment). *Chlamydia trachomatis* stabilizes host cell FAs through the action of its type III secreted effector TarP. This stabilization involves vinculin-dependent recruitment of TarP to FA sites, an increase in zyxin-positive FAs, as well as the vertical redistribution of paxillin and FAK to the force transduction layer (indicated with asterisk), an alteration whose full functional consequence remains unknown. *Yersinia pseudotuberculosis* expresses YopH, which can induce cell detachment via interaction with p130Cas and the subsequent dephosphorylation of p130Cas, paxillin and FAK.

**Table 1 ijms-22-01358-t001:** Major phosphorylation sites with assigned function present within focal adhesion (FA) proteins.

FA Protein Phosphorylation Events with an Assigned Biological Function	
Phosphorylation Site	Function
**Paxillin**—As summarized in [46]	
Tyr 31; Tyr 118; Ser 178	Regulation of Cell Migration and FA Turnover [165,166,167,168]
Ser 106; Ser 231; Ser 290	Regulation of Pax disassembly from FAs [169,170]
Ser 126; Ser 130	Translocation of Pax from FAs to the cytosol [171]
Ser 398; Ser 403; Ser 457; Ser 481	Adhesion regulation; Pax localization at FAs [172,173]
Ser 272	Regulation of Ras activity; adhesion/protrusion; inhibition of nuclear export [142]
Ser 188; Ser 190	Integrin Activation [174]
**Talin**—As mapped in [175]	
Thr 114; Thr 150	Negative regulation of integrin activation; Regulation of calpain-mediated cleavage of talin and FA turnover [176,177,178]
Thr 152	Talin recruitment to integrin adhesion sites and maintaining muscle attachment in *Drosophila* [179]
Ser 425	Inhibits binding to Smurf1—thereby preventing talin head ubiquitylation and degradation; Favors adhesion assembly, talin activity and integrin activation [98]
Ser 446	Regulation of calpain-mediated cleavage of talin and FA turnover [178]
**Vinculin**	
Tyr 100; Tyr 1065	Favors activation by promoting talin and actin binding; Regulates binding to the Arp2/3 complex; Focal adhesion development and maturation [180,181,182]
Ser 1033; Ser 1045	Favors activation by promoting talin and actin binding; Focal adhesion development and maturation [181]
**p130Cas**	
Tyr 249; Tyr 410	Binding sites for Crk SH2 domain [183]
Ser 139; Ser 437; Ser 639	Intracellular localization [184]
**Focal Adhesion Kinase (FAK)**	
Tyr 397	Major site of FAK autophosphorylation; Binding site for the SH2 domain of Src [185]
Tyr 861	Major site of phosphorylation by Src [186]
Tyr 576; Tyr 577	Promotes open FAK conformation; Facilitates scaffold and kinase functions [124]
Tyr 925	Promotes Grb2 SH2-mediated binding [187]
Tyr 194	Activation through relief of autoinhibition [188]
**Src**—As summarized in [189]	
Tyr 213	Activation [190]
Tyr 527	Autoinhibitory phosphorylation site—promotes an inactive conformation [191]
Tyr 416	Activation from autophosphorylation [192]
Ser 17	Facilitates activation of the small G protein Rap1 [126]
Thr 34; Thr 46; Ser 72	Activation of pTyrosine^527^ Src [193]
**α-actinin-4**	
Tyr 4; Tyr 31	Reduces actin binding behavior [194]
Tyr 265	Enhanced actin binding behavior; Susceptibility to calpain-mediated cleavage [195]
**Zyxin**	
Ser 142	Regulates release from autoinhibitory head: tail interaction; cell-cell adhesion regulation [127]
**RIAM**	
Tyr 340	Translocation to plasma membrane; β2 integrin designated lymphocyte functional antigen 1 (LFA-1) activation [196]
Tyr 45	Release of autoinhibitory configuration [123]

**Table 2 ijms-22-01358-t002:** Microbial modulation of FA proteins.

Microbial Adhesins That Modulate “Outside-In” Signaling				
Fibronectin				
Adhesin/Microbe	Interactions Detected	Effect on Focal Signaling/Complex/Adhesion	Microbial Advantage	References
**FlpA**/*Campylobacter jejuni*	Fibronectin: FlpA	Required for ERK 1/2 phosphorylation	Invasion	[204,205]
**CadF**/*Campylobacter jejuni*	Fibronectin: CadF	Increased paxillin phosphorylation and actin polymerization	Invasion	[204,206]
**Opc**/*Neisseria meningitidis*	OpC: serum factors	Increased phosporylation of Src and FA-associated proteins	Invasion	[207]
**SfbI**/*Streptococcus pyrogenes*	Fibronectin: SfbI	Induces integrin clustering which results in the recruitment of paxillin, FAK, and other FA proteins to site of entry; intiates FAK autophosphorylation	Invasion	[208,209]
**FnBPA-B**/*Staphylococcus aureus*	Fibronectin: FnBPA-B	Activation of integrin signalling, including FAK-Src mediated phosphorylation of cortactin which resulted in actin rearrangement and induced uptake	Invasion	[210,211]
**YadA**/Enteropathogenic *Yersinia*	Fibronectin: YadA	Triggers FAK-Src complex formation and subsequent Ras activation	Invasion	[212,213]
**Beta-integrins**				
**NadA**/*Neisseria meningitidis*	NadA: α5β1 integrin	Uncharacterized	Invasion	[214]
**Invasin**/Enteropathogenic *Yersinia*	Invasin: β1 integrins	FAK recruitment of a Src family kinase and phosphorylation of paxillin	Invasion	[215,216]
**CagL**/*Helicobacter pylori*	CagL: α5β1/αVβ6	Activates FAK and Src and triggers focal adhesion formation	Invasion	[217,218,219]
**TSA56**/*Orientia tsutsugamushi*	TSA56: α5β1 integrin	Fibronectin-binding domain of TSA56 blocked bacterial internalization	Invasion	[220]
**Tat**/HIV-1	α5β3-integrin	Increased FAK phosphorylation, RhoA and Src activation	Spread	[221]
**Secreted Effectors that modulate “inside-out” signaling**				
**p130Cas**				
**Effector/Microbe**	**Interactions detected**	**Effect on focal signaling/complex/adhesion**	**Microbial Advantage**	**References**
**YopH**/Y. *pseudotuberculosis*	YopH: p130Cas	Cas disassembles focal complexes in presence of YopH	Cell detachment/spread	[222]
	YopH PTPase activity	Dephosphorylation of p130Cas, paxillin and FAK	Cell detachment/spread	[223,224]
		Relocalization of FAK and Cas to the cytosol	Cell detachment/spread	[223]
		Dephosphorylation of of FYB, SKAP-HOM, p55	Cell detachment/spread	[225,226]
	YopH(C403A)	Fyn and p130Cas localize to FAs in presence of kinase-dead YopH	Cell detachment/spread	[225]
	YopH(Q11)	Does not bind p130Cas and is readily phagocytized	Colonization	[227]
	YopH (N-term deleted)	Decreased substrate binding and decreased virulence in IP mouse model	Colonization	[224,227]
**Paxillin**				
**BE6**/Bovine Papilloma Virus-1	BE6: Paxillin	Depolymerizes actin stress fibers	Cell transformation	[160]
	BE6: Paxillin (LD1)	Blocks paxillin interaction with vinculin and FAK	Cell transformation	[162]
**E6**/HPV-16	E6: Paxillin	LD1 domain of E6 interacts with Paxillin and decreases FA formation	Cell transformation	[161,162]
**EspC**/Enteropathogenic *E. coli*	EspC: Paxillin	Serine protease activity of EspC cleaves Paxillin	Cell detachment/spread	[107]
**Vinculin**				
**IpaA** */Shigella dysentirae*	IpaA (N-term): Vinculin	Blocks vinculin: talin interaction, vinculin binds F-actin and induces depolymerization of stress fibers	Invasion	[228]
**IpaA** */Shigella flexneri*	IpaA: Vinculin	Focal complexes form at invasion site	Invasion	[229]
	IpaA (VBS1–2): Vinculin	Recruitment of Vinculin to invasion site forms actin-vinculin cup around *Shigellae*	Invasion	[230]
	IpaA (VBS3): Vinculin	Inhibition of head and tail domains and blocks vinculin-actin association	Invasion	[231]
	IpaA: Vinculin	Vinculin-binding excludes Talin, preventing Talin: B1-integrin interaction	Cell rounding	[232]
**TarP**/*Chlamydia caviae*	TarP (VBD1–3): Vinculin	TarP binds vinculin, inducing actin polymerization at the invasion site	Invasion	[233]
**TarP**/*Chlamydia trachomatis*	TarP (LDVBD): Vinculin	TarP C-term recruitment to FAs requires vinculin, and increases FA size and stability.	Adhesion	[234]
**Sca4**/*Rickettsia rickettsii*	Sca4: Vinculin	Two VBD sites in Sca4 enable it to outcompetes alpha-catenin for vinculin binding, causing vinculin to relocalize from the cell periphery to internal sites.	Spread	[235]
**Sca4**/*Rickettsia pakerii*	Sca4: Vinculin	Sca4 interaction with vinculin blocks its ability to increase tension and barrier function at cell-cell junctions.	Spread	[236]
**Certhrax**/*Bacillus cereus*	Certhrax: Vinculin	Certhrax interacts with and ADP-ribosylates vinculin at R433	Spread	[237,238,239]
**FAK**				
**Tat**/HIV-1	Tat	Tat exposure increases phosphorylation of FAK-Y20 and increased FA numbers	Spread	[240]
**TarP**/*Chlamydia caviae*	TarP (LD): FAK	TarP orchestrates signaling through FAK-cdc42-Arp2/3 to induce actin polymerization	Invasion	[241]
**EspC**/EPEC	EspC: FAK	Serine protease activity of EspC cleaves FAK	Cell rounding	[107]
**TK (ORF21)**/KSHV	TK: FAK	Kinase activity of TK was required for relocalization of pFAK from peripheral FAs to internal sites, increasing cell detachment	Cell rounding	[242]
**ILK**				
**OspE**/*Shigella flexneri*	OspE: ILK	OspE-ILK localizes to FA, which results in increased FA size and stability, and decreased motility. OspE decreased phosphorylation of ILK targets, FAK and paxillin. OspE is required for efficient *Shigella* colonization of guinea pig colons.	Adhesion	[243,244]
**EspO1**/EHEC	EspO1: ILK	EspO1–1 localizes with ILK and prevents FA disassembly	Adhesion	[245]
**Talin**				
**Tir**/EPEC	Tir: Talin	Actin polymerization at EPEC adherence site	Invasion	[246]
**IpaA**/*Shigella flexneri*	IpaA: Talin	IpaA recruits talin to coat *Shigella* at invasion sites, and leads to the formation of extra long filopodia.	Invasion	[247]
**Zyxin**				
**E6**/HPV-6	E6: Zyxin	E6 interaction with LIM3 domain of zyxin increases its nuclear translocation.	Unknown	[159]
**Src**				
**E4orf4**/Adenovirus 2	E4:c-Src	Modulation of Src kinase activity increases phosphorylation of coractin and leads to decreased phosphorylation of FAK and paxillin.	Cell death	[109]
		E4orf4 interaction with c-Src increases phosphorylation of MLC. E4orf4-Src localize to juxtanuclear actin complex that involves Rho GTPases.	Cell death	[111]
**CagA**/*Helicobacter pylori*	CagA: Src	Src phosphorylates CagA, then phosphorylated CagA inhibits the catalytic activity of Src. Vinculin does not get phosphorylated and lamellipodia formation is reduced.	Spread	[199]
**RhoA**				
**EspM2**/EHEC	EspM2: RhoA	GEF activity of EspM2 activates RhoA-ROCK-LIMK signaling, increasing stress fiber and FA formation.	Spread	[248]
	EspM2: RhoA	EspM2 increases extrusion of cells and causes a redistribution of B1-integrin and ZO-1/tight junctions in a RhoA-dependent manner	Spread	[249]
**EspO1-2**/EHEC	EspO1-2: EspM2	Suppresses EspM2 activity, preventing RhoA activation and cell contraction	Adhesion	[245]
**EspG** and **Orf3**/ EPEC	EspG: tubulin	Causes microtubule destabilization which releases the H1-GEF and in turn activates RhoA and increases stress fiber formation.	Adhesion	[245]
**E7**/HPV-16	E7(CR3): p190RhoGAP	Interaction wtith p190RhoGAP, a RhoA inhibiting hydrolase decreased F-actin levels and cell area in transfected cells.	Adhesion	[250]

**Table 3 ijms-22-01358-t003:** Tools used in reviewed work to study FA biology.

Genetic Alterations			
FA-Associated Protein	Mutation/Deletion	Phenotype	References
RhoA	T19N	Dominant negative	[292]
RhoA	V14	Constitutively active	[293]
Src	K297M	Kinase-inactive	[207]
Src	K297M, Y529F	Dominant negative	[294]
Src	Y527F	Constitutively active	[294]
FAK	Y397F	Constitutively active; Cannot be auto-phosphorylated	[125]
FAK	D562A	Kinase-inactive	[216]
FAK	K464R	Kinase-inactive	[216]
FRNK	C-terminal domain	Dominant negative—blocks kinase activity and autophosphorylation of Y397	[295]
Talin1	L325R (FERM domain)	Compromises ability of talin1 to activate integrins without affecting binding to the NPxY motif	[296]
Talin1	R2526G	Blocks talin1 dimerization and reduces activity of C-terminal actin-binding site	[297]
Talin1	L432G	Resistant to calpain2 cleavage	[89]
Talin1	E1770A	Constitutively active; F3-R9 interaction disrupted	[117]
RIAM	E60A/D63A	Constiutively active; Disrupts inhibitory region (IN) and RA domain interaction	[123]
Vinculin	“T12” and “T12K”	Constitutively active; Destabilize vinculin head-tail interaction	[120,121]
Cortactin	W22A (NTA domain)	Cannot bind Arp2/3 complex	[298]
Cortactin	W525K (SH3 domain)	Impaired dynamin binding	[299]
Zyxin	S142D	Constitutively active; Disrupts interaction between ActA and LIM regions	[127]
α-actinin	“NEECK”	Constitutively active	[128]
**Chemical inhibitors for investigating FA signaling and dynamics**			
**Inhibitor**	**Target**	**Function inhibited**	**Reference**
Y27632	ROCK	Competes with ATP for binding to ROCK’s catalytic site	[300]
Blebbistatin	Myosin II	Inhibits myosin ATPase activity	[301]
Genistein	Protein Tyrosine Kinases	Inhibits signaling molecules within the Receptor-MAPK or Receptor-PI3K/AKT cascades	[302]
PP2	Src	Selectively inhibits Src-family kinases	[303]
SU6656	Src	Broadly inhibits Src-family kinases	[304]
PF 573882	FAK	Targets FAK catalytic activity by interaction with ATP-binding pocket; blocks Y397 phosphorylation	[305]
Cytochalaisin D	Actin	Inhibits actin polymerization by preventing actin-cofilin interaction	[306]
Latrunculin B	Actin	Inhibits actin polymerization by sequestering G-actin	[307]
Jasplakinolide	Actin	Induces actin polymerization and stabilizes F-actin	[308]
Wikostatin	N-WASP	Stabilizes N-WASP in its autoinhibited state	[309]
Leptomycin B	CRM1-mediated translocation	Inhibits the export of proteins from the nucleus to the cytoplasm	[138]
**Established cell lines**			
**Name of cell line**	**Deletion**	**Phenotype**	**Reference**
MEF	FAK	Larger and more numerous focal adhesions; Reduced cell motility	[310]
MEF/ASML	Vinculin	Reduction in directionally persistent migration and traction force generation	[311]
MEF/Mouse ES	Talin1	Unable to assemble vinculin or paxillin containing focal adhesions	[312]
MEF	Src, Yes, Fyn	Reduction in the tyrosine phosphorylation levels of FAK, p130Cas and paxillin	[313]
Mouse ES Cells	ILK	Impaired cell spreading and delayed formation of adhesions	[314]
MEF	Paxillin	Decreased cell spreading and migration; Abnormal FAs; Inefficient localization and phosphorylation of FAK	[315]
MEF	Zyxin	Reduced Ena/VASP proteins at adhesion sites; Increased motility; Deficits in actin stress fiber remodeling	[316]
MEF	p130Cas	Slow disassembly of focal adhesions at the cell front	[48]
PMNs	RIAM	Defects in adherence and cell spreading; Impaired activation of β2 integrins	[317]
GD25 (mouse)	β1-integrin	Low level of attachment to ECM substrates	[318]

## Data Availability

Not applicable.

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
