# Peer review of "Manipulation of Focal Adhesion Signaling by Pathogenic Microbes"

_ijms, 2021, doi:10.3390/ijms22031358_

Round 1

Reviewer 1 Report

This is a well-written and very thorough review on Focal adhesions within the cell and how they can be manipulated via various pathogens during the infection process. The manuscript reads well and the figures aid in understanding and descriptions.  Only minor revisions are requested by this reviewer.

Overall comments:

The introduction, while thorough, is a bit long. Removing specific examples (lines 88-96 and 115-120) would shorted the intro and remove redundancy.

Figure 4 and Table 2 should be referred to in the body of the manuscript and not just the conclusions.

Figure 4 and Table 2 should also be moved up into the body of the manuscript and away from the conclusions section.

Authors need to adjust throughout the manuscript to make sure that when the first time an organisms name is used it is expanded and subsequent times it is abbreviated. Some specific cases are noted below but they were to numerous to keep track of.

Specific comments:

Line 91: Define TarP since it is the first time used instead of line 1309

Line 96: give at least a couple examples of which pathogens were identified to have this mimicry

Lines 253, 266 272, 274: abbreviate Listeria monocytogenes

Line 696: Define FAT here instead of Line 1348

Line 714: Define PABP1

Line 897: abbreviate Campylobacter jejuni 

Line 1273: Shigella should be in italics

Line 1287: Expand R. parkerii  since this is the first use

Line 1304: Chlamydia should be in italics

Line 1307: undergoes should be undergo as Chlamydia are presumed to be plural

Line 1313: TarpVBS1 should be TarPVSB1

Line 1317: Where is pFAK phosphorylated?

Reviewer 2 Report

General comments

The review is well written and comprehensive. At times, ideas presented seem disjointed or it is unclear of the connection among concepts, but the authors do a very nice job of pulling for different areas of study and various pathogens to inform their Review. The figures are well done and provide sufficient information and detail for the reader. Overall, the authors provide sufficient details and overviews of new concepts (e.g., autoinhibition, proteolytic cleavage, phosphorylation) and refer the reader to relevant and recent reviews for more information.

Specific comments

The term pathogen is used throughout but the Review appears to focus primarily on bacterial pathogens with some viral examples included. There is no mention of fungal pathogens. Do such systems exist in fungal pathogens and with similar modes of action? If so, references to fungal pathogens exhibiting similar mechanisms should be included or if reviewed elsewhere, a statement early in the review about consciously excluding fungal pathogens should be made.

Some paragraphs seem to jump around in topic, making it difficult to focus on the main message. For example, starting at line 121, presentation of phosphorylation and dephosphorylation, degradation via proteolytic cleavage, Kindler syndrome, and cancer are all covered within a few sentences but the connection to microbial pathogens is not clear.

Line 141 – why focus on the subset listed? Why not other components?

Check proper scientific nomenclature of pathogens following initial introduction (e.g., Lines 826 & 832 Helicobacter pylori) throughout manuscript.

Italics from line 952-965 likely formatting issue. Some minor formatting issues throughout, please check.

Line 988: Opc – if this is a sub-heading, recommend formatting to highlight its placement or include sub-sub-heading. Same suggestion throughout subsequent sections.

EPEC and EHEC already introduced earlier on, not needed again (line 1104)

Table 2: text is very small. Unsure if readable for publication.

Check use of FA vs. focal adhesion throughout text
